# How rotation invariant algorithms are fooled by noise on sparse targets

**Manfred K. Warmuth**                                        MANFRED@GOOGLE.COM
*Google Inc.*

**Wojciech Kotłowski**                                  WKOTLOWSKI@CS.PUT.POZNAN.PL
*Institute of Computing Science, Poznań University of Technology, Poznań, Poland*

**Matt Jones**                                                   MCJ@COLORADO.EDU
*Google Inc. and University of Colorado Boulder, Colorado, USA*

**Ehsan Amid**                                                  EAMID@GOOGLE.COM
*Google Inc.*

**Editors:** Gautam Kamath and Po-Ling Loh

## Abstract

It is well known that rotation invariant algorithms are sub-optimal for learning sparse linear problems, when the number of examples is below the input dimension. This includes any gradient descent trained neural net with a fully-connected input layer initialized with a rotationally symmetric distribution. The simplest sparse problem is learning a single feature out of $d$ features. In that case the classification error or regression loss of rotation invariant algorithms grows with $1 - n/d$, where $n$ is the number of examples seen. These lower bounds become vacuous when the number of examples $n$ reaches the dimension $d$. After $d$ examples, the gradient space has full rank and any weight vector can be expressed, including the unit vector that determines the target feature.

In this work, we show that when noise is added to this sparse linear problem, rotation invariant algorithms are still sub-optimal after seeing $d$ or more examples. We prove this via a lower bound for the Bayes optimal algorithm on a rotationally symmetrized problem. We then prove much better upper bounds on the same problem for a large variety of algorithms that are non-invariant by rotations.

Finally, we analyze the gradient flow trajectories of many standard optimization algorithms (such as AdaGrad) on the same noisy feature learning problem, and show how they veer away from the noisy sparse targets. We then contrast them with a group of non-rotation invariant algorithms that veer towards the sparse targets.

We believe that the lower bounds method and trajectory categorization will be crucial for analyzing other families of algorithms with different classes of invariances.

**Keywords:** rotation invariance, feed forward nets, lower bounds, multiplicative updates, sparsity.

## 1. Introduction

Any gradient descent (GD) trained neural net with a fully-connected input layer is rotation invariant when initialized with a rotationally symmetric distribution. The reason is that if the input instances are rotated, then weights rotate as well and the dot products (logits) feeding into the non-linearities of the first layer remain unchanged. It is well known that rotation invariant algorithms are fundamentally inferior when learning certain sparse linear problems (Warmuth and Vishwanathan, 2005; Ng, 2004; Li et al., 2021; Warmuth et al., 2021). The simplest sparse linear problem is learning one out of $d$ features. When the instances are for example the $d$ orthogonal rows of a Hadamard matrix and a

rotation invariant algorithm is given $n \leq d$ training instances,[1] then its generalization error on all $d$ instances is at least $1 - n/d$. There are classification (Ng, 2004; Li et al., 2021) and regression versions of this problem (Warmuth and Vishwanathan, 2005; Warmuth et al., 2021), but the lower bound is essentially the same, proven with different techniques.

The lower bounds of $1 - n/d$ are contrasted by the much lower upper bounds of $O(\frac{\log d}{n})$ on the generalization error that hold for a large variety of different algorithms that distinguish between individual features and are not rotation invariant[2]: Lasso which adds an $L_1$ penalty to the loss (Candes and Tao, 2007; Rigollet and Hütter, 2023), variants of the Exponentiated Gradient updates[3] $EG^\pm$, $EGU^\pm$ (Kivinen and Warmuth, 1997), which update the feature weights by multiplicative factors, as well as GD on the 2-layer neural network of Figure 1, called *spindly* in this paper (Woodworth et al., 2019; Vaskevicius et al., 2019; Amid and Warmuth, 2020).

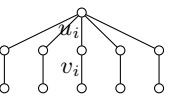

**Figure 1:** Spindly

However, the situation is much different when $n \geq d$, which is the case in most relevant settings. In particular, the lower bound of $1 - n/d$ becomes vacuous. Now any weight vector (including sparse ones) can be expressed as a linear combination of the instances. Not surprisingly, linear regression (which is rotation invariant) finds the unique consistent feature after seeing a noise-free training set of full rank.

In this paper we show that the gap between rotation invariant algorithm and ones that don't have this property remain when noise of variance $\sigma^2$ is added to the sparse target. Our first contribution is a lower bound of $\frac{d-1}{d} \frac{\sigma^2}{\sigma^2+n/d}$ on the generalization error of any rotation invariant algorithm, when the sample size $n$ (without replacement) exceeds the input dimension of the problem. Our lower bound technique creates a Bayesian setup, where the learning is presented with a randomly rotated version of the input instances. We prove a lower bound on the Bayes-optimal algorithm for this case. This means that no rotational invariant algorithm on the original problem can do better than the Bayes-optimal algorithm for a randomly rotated problem.[4] Second, we show in Appendix D that there are again simple algorithms that are not invariant by rotation, that achieve the lower generalization error of $O(\frac{\sigma^2 \log d}{n})$, when $n \geq d$ and noise of variance $\sigma^2$ is added to the sparse linear target: Variants and approximations of $EG^\pm$ including GD on the spindly network of Figure 1. We also prove the same upper bound without the $\log d$ factor for a novel *priming method* that was conjectured to learn sparse linear problems in (Warmuth and Amid, 2023) and can be seen as a special case of the Recursive Feature Machines of (Radhakrishnan et al., 2022): Find the linear least squares solution $\boldsymbol{w}$, and then after reweighing the $i$th feature with $w_i$, apply tuned ridge regression. The upper bound (with the $\log d$ factor[5]) has also been proven for Lasso (Rigollet and Hütter, 2023).

We are not interested in linear problems per se, but any fancy non-linear model should at least be able to handle the linear case. We visualize how the performance gap between rotationally invariant

---

1. Learning is harder for sampling with replacement. However, lower bounds are stronger for sampling without replacement and the boundary of getting a full rank instance set is clearer. We give bounds for both models.
2. The $L_1$ penalty is the reason the Lasso family of updates is not rotation invariant. The Exponentiated Gradient family of updates are motivated by regularizing the loss with a relative entropy (not rotation invariant). This puts the gradient of the weights in the exponent. The exponential is element-wise and therefore is not preserved under rotation (the exponential and rotation operations do not commute). Finally, in the GD updates of the spindly network of Figure 1, the individual weights $u_i$ and $v_i$ (and their gradients) are multiplied and this also makes them not rotation invariant.
3. The upper bounds for these updates follow by an on-line-to-batch conversion of the worst-case regret bounds of $O(\frac{\log d}{n})$ (See e.g. Kivinen and Warmuth (1997)). No worst-case regret bounds exist for the Lasso family of algorithms.
4. This proof technique is interesting in its own right and is different from the regression techniques used for the under-constrained case (Warmuth et al., 2021).
5. With current proof techniques, we were only able to remove the $\log d$ factor from the bound for priming.

and non-invariant algorithms arises by computing their trajectories in weight space over the course of learning. We focus on symmetry breaking due to the differences in the iterative update algorithm, while keeping the loss function fixed. All these algorithms converge to the same minimum but differ in how close they get to the sparse noisy target. The optimal early stopping point is the trajectory point that is closest to the target. Rotational symmetry can also be broken by adding a $L_1$ penalty to the loss. The trajectory is then a function of the scale factor multiplying the $L_1$ penalty (Tibshirani, 2015).

We observe that when the number of examples is larger than the input dimension and the target is sparse linear plus noise, then multiplicative updates, GD on the spindly network, priming and Lasso all produce trajectories that pass extremely close to the sparse target before finally veering away and learning the noise. Rotation invariant algorithms cannot produce such a bias, even when the input is unbalanced (anisotropic covariance) provided it is drawn from a rotationally symmetric distribution. In essence, they learn the signal and the noise at the same rate. Surprisingly, adaptive learning rate algorithms such as Adagrad (Duchi et al., 2011a) and Adam (Kingma and Ba, 2015) show the opposite bias, producing trajectories that are curved away from sparse solutions and learn the noise before the signal.

Third, we arrive at these observations by deriving closed-form solutions for the continuous time (gradient flow) weight trajectories of each algorithm for sparse noisy regression problems. We found closed forms for the isotropic covariance case but also visualize the trajectories of the anisotropic case in Appendix E. We show how the behavior of the different algorithms can be understood from a geometric analysis that recasts mirror descent and other preconditioned gradient methods as gradient descent under an altered geometry, where the preconditioner acts as a Riemannian metric.

Finally, we make some preliminary experimental observations for nonlinear models by contrasting gradient descent training on a simple three-layer feed forward neural net when a) each logit on the first layer is connected to all inputs versus b) connected to all inputs via the spindly network (Figure 1). Our experiments show in case b), gradient descent produces sparse solutions for the input layer, is much less confused by additional noisy features and can learn quickly from informative features.

## 1.1. Additional related work

There is a long history of contrasting the generalization ability of additive versus multiplicative updates (see e.g. Kivinen et al. (1997); Kivinen and Warmuth (1997)). Surprisingly, there is a connection between both update families rooted in the observation that when the weights are products of parameters, then gradient descent is biased towards sparsity (Gunasekar et al., 2017; Kerekes et al., 2021). More precisely, the gradient flow of multiplicative updates on a linear neuron equals the gradient flow of GD on the spindly network of Figure 1 (Amid and Warmuth, 2020). The strictly increasing $\text{arcsinh}(x)$ function is the link function, that defines the unnormalized two-sided Exponentiated Gradient update $\text{EGU}^\pm$ as mirror descent (Warmuth, 2007). Also, by *stretching* this link with a scale factor $\alpha \geq 1$ to $\alpha \, \text{arcsinh}(x/\alpha)$, we can interpolate between $\text{EGU}^\pm$ and GD (Ghai et al., 2020; Woodworth et al., 2020). An alternative is to introduce powers into the spindly network (Warmuth et al., 2021). For the case $n \leq d$, these updates are known to converge to a minimum norm solution that interpolate between the 1-norm and 2-norm (Woodworth et al., 2020; Warmuth et al., 2021).

In this paper we prove lower bounds for sparse noisy regression problems when $n \geq d$. It might also be possible to obtain lower bounds for classification, when the number of examples exceeds the VC dimension, building on the setup of (Ng, 2004; Li et al., 2021). However, for technical reasons,

the additional loss in the VC dimension / Rademacher complexity has a square root term, that makes this bounding methodology non-optimal in the noisy case (Srebro et al., 2010). The upper bounds for the spindly network have also been proven assuming the Restricted Isometry Property (RIP) (Vaskevicius et al., 2019). For completeness we give self-contained proofs in Appendix D.3 that also apply to variants of the multiplicative updates and priming.

Note that we have hardness results for a fixed noisy sparse linear problem: First feature plus noise. We prove that the gap in performance is due to rotation invariance. So we avoid using staircase, cork screw or cryptographically hard functions (Abbe and Boix-Adsera, 2022) for proving lower bounds. Also, in Shamir (2018), the target class is much more complicated, and the hard input distribution is algorithm-specific and therefore no fixed problem is given that is hard for all algorithms in the invariance class.

## 2. The lower bound method

Our method for proving lower bounds in Section 2.2 builds on the following observation: Given any rotation invariant algorithm and any learning problem, the algorithm will achieve the same loss on all rotated versions of that problem. We can therefore consider a Bayesian setting where the problem is sampled uniformly from all rotated versions, and the optimal solution provides a lower bound on the loss of the algorithm. Intuitively, being rotation invariant forces an algorithm to be agnostic over all possible rotations of the problem, and hedging its bets in this way prevents it from excelling at any specific problem instance. In Section 2.3 we apply this reasoning to linear regression to show that a rotation invariant algorithm cannot efficiently learn sparse solutions, because it must be equally efficient at finding any other solution (including rotated, non-sparse ones).

### 2.1. Rotation invariance and problem setup

An *example* $(\boldsymbol{x}, y)$ is a $d$-dimensional vector $\boldsymbol{x}$, followed by a real-valued label $y \in \mathbb{R}$. We specify a training set as a tuple $(\underset{n,d}{\boldsymbol{X}}, \underset{n}{\boldsymbol{y}})$ containing $n$ training examples, where the rows of *input matrix* $\boldsymbol{X}$ are the $n$ (transposed) input vectors and the *target* $\boldsymbol{y}$ is a vector of their labels.

A *learning algorithm* is a mapping which, given the training set $(\boldsymbol{X}, \boldsymbol{y})$, produces a real-valued *prediction function* $\mathbb{R}^d \ni \boldsymbol{x} \mapsto \widehat{y}(\boldsymbol{x}|\boldsymbol{X}, \boldsymbol{y}) \in \mathbb{R}$. An algorithm is called *rotation invariant* (Warmuth and Vishwanathan, 2005; Warmuth et al., 2021) if for any orthogonal matrix $\underset{d,d}{\boldsymbol{U}}$ and any input $\boldsymbol{x} \in \mathbb{R}^d$:

$$\widehat{y}(\boldsymbol{U}\boldsymbol{x}\,|\,\boldsymbol{X}\boldsymbol{U}^\top, \boldsymbol{y}) = \widehat{y}(\boldsymbol{x}\,|\,\boldsymbol{X}, \boldsymbol{y}). \tag{1}$$

In other words, the prediction $\widehat{y}(\boldsymbol{x}|\boldsymbol{X}, \boldsymbol{y})$ remains the same, if we rotate both $\boldsymbol{x}$ and all examples from $\boldsymbol{X}$ by the same orthogonal matrix $\boldsymbol{U}$. If the algorithm is randomized, based on an internal random variable[6] $Z$, then $\widehat{y}(\boldsymbol{x}|\boldsymbol{X}, \boldsymbol{y})$ is a random variable given by some function $f_{\boldsymbol{x},\boldsymbol{X},\boldsymbol{y}}(Z)$, and the equality sign in equation (1) should be interpreted as *identically distributed.*

Our lower bounds in Sections 2.2-2.3 hold for any rotation invariant algorithm. In particular, Warmuth et al. (2021) have shown that any neural network with a fully-connected input layer (and arbitrary remaining layers), in which the weights in the input layer are initialized with a rotation invariant distribution (e.g. i.i.d. Gaussians) and are trained by GD, is rotation invariant and is thus subject to our lower bound. The reason is that such a network can be written as $f(\boldsymbol{w}_1 \cdot \boldsymbol{x}, \boldsymbol{w}_2 \cdot \boldsymbol{x}, \dots, \boldsymbol{w}_h \cdot \boldsymbol{x}, \boldsymbol{\theta})$, and the gradient $\nabla_{\boldsymbol{w}_i} f$ for unit $i$ in the first hidden layer is equal

---

6. For example, in neural networks $Z$ would correspond to a random initialization of the parameter vector.

to the instance $\boldsymbol{x}$ times a scalar that depends on $\boldsymbol{x}$ only via $\boldsymbol{w}_i \cdot \boldsymbol{x}$. Also, updating the weights $\boldsymbol{\theta}$ of the later layers depends on the input only via the $\boldsymbol{w}_i \cdot \boldsymbol{x}$. Therefore, it is easy to show by induction on the time step $t$, that rotating all instances results in the same rotation of all $\boldsymbol{w}_{i,t}$. The rotation has no effect on the dot products $\boldsymbol{x} \cdot \boldsymbol{w}_i$ computed at the input layer. In contrast, learning with $f((\boldsymbol{u}_1 \odot \boldsymbol{v}_1) \cdot \boldsymbol{x}, \dots, (\boldsymbol{u}_h \odot \boldsymbol{v}_h) \cdot \boldsymbol{x}, \boldsymbol{\theta})$, where the parameters $\boldsymbol{w}_i$ connected to the input $\boldsymbol{x}$ are replaced by $\boldsymbol{u}_i \odot \boldsymbol{v}_i$ (spindlified), is not invariant by rotation.

Formally, a learning problem is defined by: (a) an input distribution $p_{\text{in}}(\tilde{\boldsymbol{X}})$ with the input matrix $\underset{n+1,d}{\tilde{\boldsymbol{X}}} = [\boldsymbol{X}; \boldsymbol{x}_{\text{te}}^{\top}]$ consisting of the training matrix $\boldsymbol{X}$ plus a test instance $\boldsymbol{x}_{\text{te}}$; (b) an observation model $q(\tilde{\boldsymbol{y}}|\tilde{\boldsymbol{X}})$, which gives the joint conditional distribution over $n$ training outcomes and a test outcome, $\underset{n+1}{\tilde{\boldsymbol{y}}} = [\boldsymbol{y}; y_{\text{te}}]$, (c) a loss function $\mathcal{L}(\hat{y}, y)$. We assume the input distribution is rotationally symmetric, meaning $p_{\text{in}}(\tilde{\boldsymbol{X}}) = p_{\text{in}}(\tilde{\boldsymbol{X}}\boldsymbol{U}^{\top})$ for any orthogonal $\boldsymbol{U}$. The task of any algorithm will be to produce predictions $\hat{y}(\boldsymbol{x}_{\text{te}}|\boldsymbol{X}, \boldsymbol{y})$ to minimize the loss on the test outcomes, $\mathcal{L}(\hat{y}, y_{\text{te}})$. Note that this setup allows arbitrary conditional dependencies among observations (not just i.i.d problems), including dependencies between the training and the test sets.

## 2.2. Lower bounds for rotation invariant algorithms

For any orthogonal $\underset{d,d}{\boldsymbol{U}}$, define the rotated observation model as $q_{\boldsymbol{U}}(\tilde{\boldsymbol{y}}|\tilde{\boldsymbol{X}}) = q(\tilde{\boldsymbol{y}}|\tilde{\boldsymbol{X}}\boldsymbol{U}^{\top})$. Now define a new learning problem by first sampling $\boldsymbol{U}$ uniformly (under the Haar measure $p_{\text{H}}$) and then generating observations according to $q_{\boldsymbol{U}}$. This is equivalent to a symmetrized observation model $\mathring{q}$ that is a mixture over all $q_{\boldsymbol{U}}$:

$$\mathring{q}(\tilde{\boldsymbol{y}}|\tilde{\boldsymbol{X}}) = \int q_{\boldsymbol{U}}(\tilde{\boldsymbol{y}}|\tilde{\boldsymbol{X}}) \mathrm{d}p_{\text{H}}(\boldsymbol{U})$$

The Bayes optimal prediction can be expressed by computing a posterior over $\boldsymbol{U}$ and integrating expected loss over this posterior:

$$\hat{y}^{\star}(\boldsymbol{x}_{\text{te}}|\boldsymbol{X}, \boldsymbol{y}) = \arg\min_{\hat{y}} \int \mathbb{E}_{y_{\text{te}} \sim q_{\boldsymbol{U}}(\cdot|\tilde{\boldsymbol{X}}, \boldsymbol{y})}[\mathcal{L}(\hat{y}, y_{\text{te}})] p(\boldsymbol{U}|\tilde{\boldsymbol{X}}, \boldsymbol{y}) \mathrm{d}\boldsymbol{U}.$$

Thus $\mathring{q}$ is difficult, especially for large $d$, because equal prior probability must be given to all possible rotations. We define the optimal expected loss on this problem as

$$L_{\mathcal{B}}(\mathring{q}) = \mathbb{E}_{\tilde{\boldsymbol{X}} \sim p_{\text{in}}, \tilde{\boldsymbol{y}} \sim \mathring{q}(\cdot|\tilde{\boldsymbol{X}})}[\mathcal{L}(\hat{y}^{\star}(\boldsymbol{x}_{\text{te}}|\boldsymbol{X}, \boldsymbol{y}), y_{\text{te}})].$$

Our first result is that the performance of any rotation invariant algorithm on the original problem, defined by $q$, is lower bounded by $L_{\mathcal{B}}(\mathring{q})$.

**Theorem 1** *Given a rotationally symmetric $p_{in}(\tilde{\boldsymbol{X}})$, an observation model $q(\tilde{\boldsymbol{y}}|\tilde{\boldsymbol{X}})$, a loss function $\mathcal{L}$, and a rotationally invariant learning algorithm $\hat{y}_n(\cdot|\boldsymbol{X}, \boldsymbol{y})$, define the expected loss*

$$L_{\hat{y}}(q) = \mathbb{E}_{\tilde{\boldsymbol{X}} \sim p_{in}, \tilde{\boldsymbol{y}} \sim q(\cdot|\tilde{\boldsymbol{X}}), Z}[\mathcal{L}(\hat{y}(\boldsymbol{x}_{te}|\boldsymbol{X}, \boldsymbol{y}), y_{te})].$$

*This loss is bounded by $L_{\hat{y}}(q) \geq L_{\mathcal{B}}(\mathring{q})$.*

The proof in Appendix A shows any rotationally invariant algorithm will satisfy $L_{\widehat{y}}(q_U) = L_{\widehat{y}}(q)$ for all $U$. This implies $L_{\widehat{y}}(q) = L_{\widehat{y}}(\mathring{q}) \geq L_{\mathcal{B}}(\mathring{q})$.

As we show in this paper, a consequence of Theorem 1 is that rotational invariance prevents efficient learning of problems characterized by properties that are not rotationally invariant, such as sparsity. Algorithms with inductive biases for such properties are necessarily not rotation invariant and can give dramatically better performance. Although we have stated the theorem in terms of rotational invariance, it is easily extended to other transformation groups $\mathcal{T}$ on the input (by replacing $U$ with elements of $\mathcal{T}$ and requiring $p_{\text{in}}$ to be symmetric under $\mathcal{T}$). For example, natural gradient descent (NGD) is often touted for being invariant to arbitrary smooth reparameterization (Amari and Douglas, 1998), but this invariance comes at a cost of being unable to efficiently learn in environments that are not invariant in this way. In particular, since spindly and fully connected networks are related by smooth reparameterization, NGD performs equivalently on both (see discussion in (Kerekes et al., 2021)). Thus the lower bounds we prove apply to NGD on the network in Figure 1, whereas we show vanilla gradient descent on this network breaks the lower bound.

### 2.3. Lower bound for least-squares regression

We demonstrate how Theorem 1 can provide a quantitative lower bound for a specific class of learning problems. We then show in Appendix D, how this bound is easily beaten by algorithms that are non-invariant by rotation. In the specific problem class we consider, the number of training examples is $n = md$ for some integer $m$, and $X$ consists of $m$ stacked copies of a matrix $H = \sqrt{d}\underset{d,d}{V}$ (i.e. $X = \underset{\times m}{[H;\ldots;H]}$), where $V$ is a random orthogonal matrix distributed according to the Haar measure.[7] Thus $p_{\text{in}}(X)$ is rotationally symmetric. The test input is one of the rows of $H$, $x_{\text{te}} = h_k$, with index $k$ drawn uniformly at random. We assume the labels are the first feature of $\tilde{X}$ plus Gaussian noise, that is

$$q(\tilde{y}|\tilde{X}) = \mathcal{N}(\tilde{y}|\tilde{X}e_1, \sigma^2 I_{n+1}), \qquad \text{where } e_1 = (1, 0, \ldots, 0)^\top, \tag{2}$$

which can be be equivalently written as

$$y = Xe_1 + \xi, \text{ where } \xi \sim N(0, \sigma^2 I_{md}), \text{ and } y_{\text{te}} = h_k^\top e_1 + \xi_{\text{te}}, \text{ where } \xi_{\text{te}} \sim N(0, \sigma^2).$$

Note that while the inputs are shared in the training and the test parts, the test label is generated using a fresh copy of the noise variable $\xi_{\text{te}}$. The fixed choice of $e_1$ as opposed to any other $e_i$ is made w.l.o.g. Indeed, Theorem 1 implies that a rotationally invariant algorithm will have the same loss for $Xe_1$ as it will for $Xw$ for any other unit vector $w$.

The accuracy of prediction $\widehat{y} = \widehat{y}(\cdot|X, y)$ on the test example $(h_k, y_{\text{te}})$ is measured by the *squared loss* $\mathcal{L}(\widehat{y}, y_{\text{te}}) = (\widehat{y} - y_{\text{te}})^2$. Let $\widehat{y}$ denote the vector of predictions for all possible test instances, $\widehat{y}_k = \widehat{y}(h_k|X, y)$, and let $y_{\text{te}}$ denote the vector of test labels, $y_{\text{te},k} = h_k^\top e_1 + \xi_{\text{te}}$, which can be jointly written as $y_{\text{te}} = He_1 + \xi_{\text{te}}\mathbf{1}$ with $\mathbf{1} = (1, \ldots, 1)$. The expected value of the loss over

---

7. We multiply $V$ by $\sqrt{d}$ in order to keep the lengths of examples $\|x_i\|$ (rows of $X$) equal to $\sqrt{d}$, so that their coordinates $x_{ij}$ (individual features) are of order 1 on average; any other scaling would work as well, resulting only in a relative change of the effective label noise level.

the random choice of $k \in \{1, \ldots, d\}$ and over the independent test label noise is given by:

$$
\begin{aligned}
\mathbb{E}_{k,\xi_{\text{te}}}[\mathcal{L}(\widehat{y}, y_{\text{te}})] &= \frac{1}{d}\, \mathbb{E}_{\xi_{\text{te}}}\big[\|\widehat{\boldsymbol{y}} - \boldsymbol{y}_{\text{te}}\|^2\big] = \frac{1}{d}\, \mathbb{E}_{\xi_{\text{te}}}\big[\|\widehat{\boldsymbol{y}} - \boldsymbol{H}\boldsymbol{e}_1 + \xi_{\text{te}}\boldsymbol{1}\|^2\big] \\
&= \frac{1}{d}\|\widehat{\boldsymbol{y}} - \boldsymbol{H}\boldsymbol{e}_1\|^2 + \frac{2}{d}\underbrace{\mathbb{E}_{\xi_{\text{te}}}[\xi_{\text{te}}]}_{=0}(\widehat{\boldsymbol{y}} - \boldsymbol{H}\boldsymbol{e}_1)^\top \boldsymbol{1} + \frac{1}{d}\, \mathbb{E}_{\boldsymbol{\xi}_{\text{te}}}\underbrace{\big[\xi_{\text{te}}^2\big]}_{=\sigma^2}\|\boldsymbol{1}\|^2 \\
&= \frac{1}{d}\|\widehat{\boldsymbol{y}} - \boldsymbol{H}\boldsymbol{e}_1\|^2 + \sigma^2.
\end{aligned}
$$

Clearly, the expression above is minimized by setting the prediction vector to $\widehat{\boldsymbol{y}}^\star = \boldsymbol{H}\boldsymbol{e}_1$, and thus the smallest achievable expected loss is equal to $\sigma^2$. Subtracting this loss, we get the expression for the excess risk of the learning algorithm, which we call the *error* of $\widehat{\boldsymbol{y}}$:

$$
e(\widehat{\boldsymbol{y}}) = \mathbb{E}_{k,\xi_{\text{te}}}[\mathcal{L}(\widehat{y}, y_{\text{te}})] - \mathbb{E}_{k,\xi_{\text{te}}}[\mathcal{L}(\widehat{y}^\star, y_{\text{te}})] = \frac{1}{d}\|\widehat{\boldsymbol{y}} - \boldsymbol{H}\boldsymbol{e}_1\|^2.
$$

When the prediction is *linear*, $\widehat{\boldsymbol{y}} = \boldsymbol{H}\widehat{\boldsymbol{w}}$ for some weight vector $\widehat{\boldsymbol{w}} \in \mathbb{R}^d$, we can also refer to the error of $\widehat{\boldsymbol{w}}$ as the error of its predictions:

$$
e(\widehat{\boldsymbol{w}}) = \frac{1}{d}\|\boldsymbol{H}\widehat{\boldsymbol{w}} - \boldsymbol{H}\boldsymbol{e}_1\|^2 = \frac{1}{d}(\widehat{\boldsymbol{w}} - \boldsymbol{e}_1)^\top \overbrace{\boldsymbol{H}^\top \boldsymbol{H}}^{d\boldsymbol{I}}(\widehat{\boldsymbol{w}} - \boldsymbol{e}_1) = \|\widehat{\boldsymbol{w}} - \boldsymbol{e}_1\|^2. \tag{3}
$$

We prove the following lower bound for rotationally invariant algorithms on this problem:

**Theorem 2** *Let $\boldsymbol{V}$ be a random orthogonal matrix, and let $\boldsymbol{H} = \sqrt{d}\boldsymbol{V}$. Let $(\boldsymbol{X}, \boldsymbol{y})$ be the training set with $\underset{md,d}{\boldsymbol{X}} = [\boldsymbol{H}; \ldots; \boldsymbol{H}]$ and labels $\boldsymbol{y}$ generated according to (2). Then the expected error (with respect to $\boldsymbol{V}$) of any rotation-invariant learning algorithm is at least*

$$
\mathbb{E}_{\boldsymbol{V}}[e(\widehat{\boldsymbol{y}})] \geq \frac{d-1}{d}\frac{\sigma^2}{\sigma^2 + m}.
$$

The proof in Appendix B uses Theorem 1 and closely follows Section 2.2: We start with a Bayesian setting with the rotated observation model $q(\tilde{\boldsymbol{y}}|\tilde{\boldsymbol{X}}\boldsymbol{U}^\top)$ for a random orthogonal matrix $\boldsymbol{U}$. This is equivalent to simply rotating the target weight vector by $\boldsymbol{U}^\top$, because $\tilde{\boldsymbol{X}}\boldsymbol{U}^\top\boldsymbol{e}_1 = \tilde{\boldsymbol{X}}(\boldsymbol{U}^\top\boldsymbol{e}_1)$. Thus we can equivalently consider a linear model $\tilde{\boldsymbol{y}} = \tilde{\boldsymbol{X}}\boldsymbol{w} + \tilde{\boldsymbol{\xi}}$, where $\boldsymbol{w}$ is drawn uniformly from the unit sphere $\mathcal{S}^{d-1} = \{\boldsymbol{w} \in \mathbb{R}^d : \|\boldsymbol{w}\| = 1\}$. Given the square loss function and linear observation model, the optimal Bayes predictor is based on the posterior mean, $\mathbb{E}[\boldsymbol{w}|\boldsymbol{X}, \boldsymbol{y}]$.

Even though the posterior mean does not have a simple analytic form for the prior distribution over a unit sphere, we use results from (Marchand, 1993; Dicker, 2016) to show that the Ridge Regression (RR) predictor (which is the Bayes predictor for the Gaussian prior) with appropriately chosen regularization constant has expected error at most $\frac{1}{d}\frac{\sigma^2}{\sigma^2 + m}$ larger than that of the Bayes predictor. Thus, it suffices to analyze the expected error of the RR predictor which we show to be at least $\frac{\sigma^2}{\sigma^2 + m}$. This implies the Bayes error is at least $\frac{d-1}{d}\frac{\sigma^2}{\sigma^2 + m}$, and no other algorithm can achieve any better error for this problem. Finally, due to the rotation-symmetric distribution of the inputs, we can now apply Theorem 1, which implies that every rotation invariant algorithm has error at least $\frac{d-1}{d}\frac{\sigma^2}{\sigma^2 + m}$ for the original sparse linear regression problem $\tilde{\boldsymbol{y}} = \tilde{\boldsymbol{X}}\boldsymbol{e}_1 + \tilde{\boldsymbol{\xi}}$. For the sake of completeness, we also give an i.i.d. version of this lower bound in Appendix C.

Previous lower bound proofs for sparse problems for the case when the number of examples is less than the input dimension used fixed choices for input matrix $\boldsymbol{X}$ such as the $d$-dimensional Hadamard matrix (Kivinen et al., 1997; Warmuth and Vishwanathan, 2005). In the over-constrained case, the lower bound does not hold for any fixed choice of $\boldsymbol{X}$ (of full rank). For any fixed full-rank

$\boldsymbol{X}$, there exists a row vector $\boldsymbol{v}$ s.t. $\boldsymbol{v}\boldsymbol{X} = \boldsymbol{e}_1^\top$, and the linear algorithm $\widehat{y}(\boldsymbol{x}|\boldsymbol{X},\boldsymbol{y}) = \boldsymbol{v}\boldsymbol{X}\,\boldsymbol{x}$ achieves minimal loss $\sigma^2$ while being trivially rotationally invariant because $\boldsymbol{v}\boldsymbol{X}\boldsymbol{U}^\top\boldsymbol{U}\boldsymbol{x} = \boldsymbol{v}\boldsymbol{X}\,\boldsymbol{x}$. Thus the assumption of rotationally symmetric input distribution is essential.[8]

This counterexample requires knowledge of the order of the examples (i.e., the algorithm makes different predictions if the rows of $\boldsymbol{X}$ are permuted). Now consider $\boldsymbol{X} = [\boldsymbol{H}; \ldots; \boldsymbol{H}]\,\mathrm{diag}(\boldsymbol{p})$ with $p_1 = 2, p_{i>1} = 1$. Then the target $\boldsymbol{e}_1$ is embedded as the first principal component $\mathbb{PC}_1(\boldsymbol{X}^\top\boldsymbol{X})$, so an algorithm that used this as its weight vector, $\widehat{y}(\boldsymbol{x}|\boldsymbol{X},\boldsymbol{y}) = \mathbb{PC}_1(\boldsymbol{X}^\top\boldsymbol{X})^\top\boldsymbol{x}$, would achieve minimal loss. This algorithm is rotationally invariant but fails if the input matrix $\boldsymbol{X}$ was rotated.

## 3. Gradient flow trajectories

In our setting of overconstrained optimization without explicit regularization, all algorithms operate on the same loss landscape but follow different paths to the minimum. To understand these differences, we derive exact expressions for the weight trajectories for the continuous time (gradient flow) versions of several representative algorithms. We then use these results to analyze the algorithms' ability to learn sparse targets without overfitting to noise.

### 3.1. Preconditioning, mirror descent, reparameterization, and Riemannian descent

Gradient descent (GD) on a given loss function can be generalized in several ways, including preconditioned GD, mirror descent (MD), GD on reparameterized variables, and Riemannian GD. We show how how to translate among these formulations in the continuous-time case and express them all as preconditioned GD. Even though this common reformulation as GD might be known to some readers, we formalize it as a theorem and give a proof in an appendix.

Reparameterizing refers to defining some injective function $\widehat{\boldsymbol{w}} = g(\boldsymbol{w})$ and performing GD on the new variable: $\frac{\mathrm{d}}{\mathrm{d}t}\widehat{\boldsymbol{w}} = -\nabla_{\widehat{\boldsymbol{w}}}L$. MD (Nemirovsky and Yudin, 1983) involves an invertible mirror map $\widetilde{\boldsymbol{w}} = f(\boldsymbol{w})$ where $f$ is the gradient of a convex function, with the update $\frac{\mathrm{d}}{\mathrm{d}t}\widetilde{\boldsymbol{w}} = -\nabla_{\boldsymbol{w}}L$. Preconditioning premultiplies the gradient with a matrix $\boldsymbol{P}$ (which can depend on current weights or training history), giving the update $\frac{\mathrm{d}}{\mathrm{d}t}\boldsymbol{w} = -\boldsymbol{P}\,\nabla_{\boldsymbol{w}}L$. Explicit preconditioning methods include adaptive learning rate algorithms (Duchi et al., 2011b), natural gradient descent (Amari, 1998), and variational Bayesian methods (Lambert et al., 2022; Chang et al., 2023).

Riemannian GD generalizes to parameter spaces with non-Euclidean geometry (specifically, differentiable manifolds) (Bonnabel, 2013). The update $\frac{\mathrm{d}}{\mathrm{d}t}\boldsymbol{w}$ lies in the tangent space $T_{\boldsymbol{w}}$ (a direction of motion based at $\boldsymbol{w}$) whereas the gradient $\nabla_{\boldsymbol{w}}L$ lies in the cotangent space $T_{\boldsymbol{w}}^*$ (the dual of $T_{\boldsymbol{w}}$). Unlike in Euclidean geometry, $T_{\boldsymbol{w}}$ and $T_{\boldsymbol{w}}^*$ cannot be identified and instead mapping between them requires a metric. The metric is a linear map $\boldsymbol{\Gamma}_{\boldsymbol{w}} : T_{\boldsymbol{w}} \to T_{\boldsymbol{w}}^*$ or equivalently a bilinear map $\boldsymbol{\Gamma}_{\boldsymbol{w}} : T_{\boldsymbol{w}} \times T_{\boldsymbol{w}} \to \mathbb{R}$ which can be expressed as a matrix in $\mathbb{R}^{d\times d}$. This leads to an update that is the direction of steepest descent with respect to the geometry: $\frac{\mathrm{d}}{\mathrm{d}t}\boldsymbol{w} = -\boldsymbol{\Gamma}_{\boldsymbol{w}}^{-1}\nabla_{\boldsymbol{w}}L$.

**Theorem 3** *The continuous-time versions of reparameterized GD, MD, and Riemannian GD are all equivalent to preconditioned GD under the relation*

$$\boldsymbol{P} = \frac{\partial\boldsymbol{w}}{\partial\widehat{\boldsymbol{w}}}\left(\frac{\partial\boldsymbol{w}}{\partial\widehat{\boldsymbol{w}}}\right)^\top = \frac{\partial\boldsymbol{w}}{\partial\widetilde{\boldsymbol{w}}} = \boldsymbol{\Gamma}_{\boldsymbol{w}}^{-1}.$$

---

8. In the Hadamard matrix based lower bounds of Warmuth and Vishwanathan (2005), the ease of learning say $\boldsymbol{e}_1$ is overcome by averaging over all $n$ targets $\boldsymbol{e}_i$.

Appendix D.5 gives the proof and then uses this result (a) to derive a close connection between EGU and the spindly network and (b) to show how EGU implicitly operates on a non-Euclidean geometry (Figure 4) that tends to hold it in sparse regions of the parameter space. Rotationally invariant algorithms cannot do this because their implicit geometry must be rotationally symmetric.

## 3.2. Trajectory solutions

We derive analytic trajectories for the simple linear regression problem of Section 2.3 for the continuous-time versions of EGU, EGU$^\pm$, primed GD, and Adagrad (Duchi et al., 2011a). We give the expressions for the trajectories here and defer the derivations and full details to Appendix E.

For continuous EGU, the trajectory is (with $c_i$ a constant depending on initial conditions):

$$w_i(t) = \tfrac{1}{2} w_i^{\text{LS}} \left( 1 + \tanh \left( w_i^{\text{LS}} t + c_i \right) \right). \tag{4}$$

For continuous EGU$^\pm$, the trajectory is (with $\tau_i$ a transform of $t$):

$$w_i(t) = \frac{w_i^{\text{LS}} \sinh \tau_i + 1}{\sinh \tau_i - w_i^{\text{LS}}}. \tag{5}$$

For primed gradient flow, which is similar to the priming method analyzed elsewhere in this paper except the second stage uses gradient flow instead of ridge regression, the trajectory is

$$w_i(t) = w_i^{\text{LS}} + \left( w_i(0) - w_i^{\text{LS}} \right) e^{-2t \left( w_i^{\text{LS}} \right)^2}. \tag{6}$$

For continuous-time Adagrad the trajectory is (with constants $k_i, \ell_i$ depending on initial conditions):

$$w_i(t) = w_i^{\text{LS}} - \text{sign} \left( w_i^{\text{LS}} - w_i(0) \right) \sqrt{-\frac{8}{\beta k_i} W \left( -e^{-k_i(t+\ell_i)} \right)}, \tag{7}$$

with Lambert's W function defined by $x = W(x) e^{W(x)}$.

## 3.3. Experimental visualization

The analytic trajectories derived above are visualized in Figure 2 for 2d and 256d versions of our noisy regression problem, using both a sparse target $\boldsymbol{w} = \boldsymbol{e}_1$ and a dense (diagonal) target $\boldsymbol{w} = \boldsymbol{1}/\sqrt{d}$. All the algorithms converge to the same Linear Least Squares (LLS) solution (black dot), but they follow qualitatively different trajectories that also differ between sparse and dense targets.

Consider first the 2d problem in Figures 2(a),(d). We plot the evolution of the 2d weight vector starting at the origin and converging to $\boldsymbol{w}_{\text{LLS}}$ (with time as an implicit parameter of each curve). In Figure 2(a), priming and EGU$^\pm$ pass very close to the sparse target (red dot) and Lasso's piecewise-linear trajectory also comes close.[9] GD follows a straight line to the LLS solution, and other rotationally invariant algorithms such as ridge regression also follow this path. Surprisingly, Adagrad veers away from the sparse target. In Figure 2(d), GD again goes straight to the LLS solution, but now Adagrad curves toward the target while priming, EGU$^\pm$ and Lasso all veer away from it.

---

9. In Theorem 7, we also prove exponential convergence of the error of EG$^\pm$ in one batch update. This bound is new. Indeed using learning rate $\eta = 200$, EG$^\pm$ achieves error below $10^{-300}$ (not shown). However EG$^\pm$ makes use of the a priori knowledge of the norm of the true weight vector (which may be unreasonable).

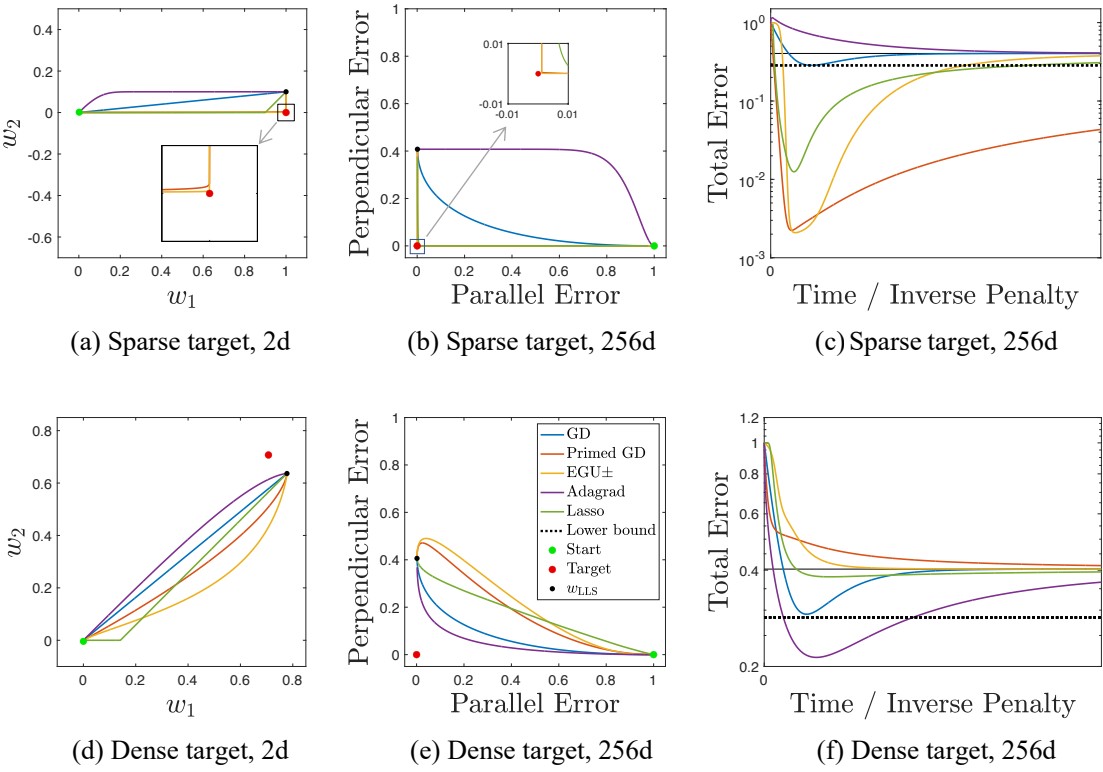

(a) Sparse target, 2d      (b) Sparse target, 256d      (c) Sparse target, 256d

(d) Dense target, 2d      (e) Dense target, 256d      (f) Dense target, 256d

**Figure 2:** Trajectories for key algorithms in 2d and 256d regression. Targets (red dots) are sparse $(1,0,\ldots,0)$ or dense $(1/\sqrt{d},\ldots,1/\sqrt{d})$. Input $\boldsymbol{X}$ is $m$ copies of the Hadamard matrix yielding $md$ training examples in every run with $m = 10$. All algorithms converge to LLS solution (black dots) but differ in the order of learning the signal versus noise. (a,d) Trajectories in 2d weight space. Noise was hand coded for a single run to illustrate differences among algorithms. (b,e) 256d trajectories, visualized as squared error parallel ($e_{\text{signal}}$) vs. perpendicular ($e_{\text{noise}}$) to the target, with total error $e = e_{\text{signal}} + e_{\text{noise}}$. Average of 10 replications per problem with observation noise $\sigma = 2$. Black dots here show average error of LLS solution across replications. (c,f) Average total error over same 10 replications for each problem. Horizontal axis scaling is arbitrary and hand chosen for each algorithm to ease comparisons. Lower bound for rotationally invariant algorithms (dotted line) is from Thm. 2. Thin black line is the mean error of the LLS solution to which all algorithms converge.

The same pattern is seen for 256d in Figures 2(b),(e). We define squared error in the direction of the target, $e_{\text{signal}}(\widehat{\boldsymbol{w}}) = (1 - \widehat{\boldsymbol{w}}^\top \boldsymbol{w})^2$, and in all perpendicular (noise) directions, $e_{\text{noise}}(\widehat{\boldsymbol{w}}) = \|(\boldsymbol{I} - \boldsymbol{w}\boldsymbol{w}^\top)\widehat{\boldsymbol{w}}\|^2$. These sum to the total error: $e_{\text{signal}}(\widehat{\boldsymbol{w}}) + e_{\text{noise}}(\widehat{\boldsymbol{w}}) = \|\widehat{\boldsymbol{w}} - \boldsymbol{w}\|^2 = e(\widehat{\boldsymbol{w}})$ (note $\|\boldsymbol{w}\|^2 = 1$). This strategy lets us visualize the high-dimensional trajectories by collapsing all dimensions perpendicular to the target into one axis. All algorithms start at $\widehat{\boldsymbol{w}} = \boldsymbol{0}$ which corresponds to $e_{\text{signal}} = 1$ and $e_{\text{noise}} = 0$. For the sparse target (Figure 2(b)), priming, EGU$^\pm$ and Lasso all learn the signal almost entirely before beginning to learn the noise.[10] GD follows a straight path that learns signal and noise at equal rates (seen as a parabola since error is square), and Adagrad learns the noise almost entirely before significantly learning the signal. For the dense target (Figure 2(e)), this

---

10. By gradually stretching EGU$^\pm$'s $arcsinh$ link as described in Section 1.1, the EGU$^\pm$ trajectories move away from the sparse targets in Figure 2(a),(b) until they morve into the GD trajectories (not shown).

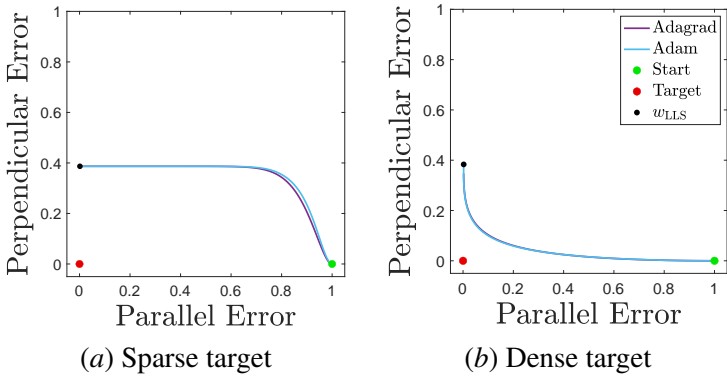

(a) Sparse target          (b) Dense target

**Figure 3:** Comparison between Adagrad (exact solution for continuous time) and Adam with discrete updates. Parameters for Adam are $\eta = .001, \beta_1 = .9, \beta_2 = .999, \epsilon = 10^{-8}$. Other details match Figures 2b and 2e. Adam's behavior is very close to Adagrad's, learning inefficiently with a sparse target (noise before signal) and efficiently with a dense target (signal before noise).

pattern is reversed, with Adagrad learning the signal faster than the noise and priming, EGU$^\pm$ and Lasso learning the noise faster than the signal (relative to GD and other rotation invariant algorithms). Notably, the differences among the algorithms are stronger in the sparse case than in the dense case.

These differences in trajectory have dramatic implications for the excess test loss as shown in Figures 2(c),(f), which show $e(\widehat{\boldsymbol{w}})$ as a function of time. In the sparse case, priming, EGU$^\pm$ and Lasso achieve minimum loss close to zero.[11] GD shows a smaller dip reflecting early-stopping regularization and comes very close to the theoretical lower bound of Theorem 2. Adagrad shows no dip at all meaning its performance is best in the LLS limit. In the dense case, the pattern is reversed, though again the differences among algorithms are smaller than in the sparse case.

The surprising behavior of Adagrad can be explained by its weight-specific adaptive learning rates. For smooth loss landscapes where gradients do not rapidly change, Adagrad's normalization term causes all weights to change at nearly the same rate. This property also applies to other adaptive optimizers such as Adam, as illustrated in Figure 3.[12] Consequently the trajectory starts along a diagonal in weight space. This inductive bias is advantageous for a dense target and disadvantageous for a sparse target. Note that this behavior is specific to methods with diagonal preconditioners; full-matrix Adagrad is rotationally invariant and does not have this inductive bias.

Finally, the lower bound of Sections 2.3, the upper bounds for Appendix D and the curves of Figure 2 assume the instance matrix $\boldsymbol{X}$ comprises $m$ copies of a scaled $d$-dimensional rotation matrix. This unusual setup was needed for technical reasons. However, if we sample the instance matrix $\boldsymbol{X}_{md,d}$ with i.i.d. Gaussian elements, then the curves remain essentially unchanged (not shown).

## 4. Noise experiments on Fashion MNIST

The assumptions in our theorems, of a rotationally symmetric input distribution and a target defined by a single feature, are too strong for real datasets. Therefore, it is important to complement our

---

11. Although not apparent, Lasso's curves in 256d are piecewise linear as they are in 2d.

12. We simulate Adam in its standard discrete time form because the PDE for the continuous version does not appear to be solvable. Nevertheless the behavior is very close to that of continuous Adagrad.

theoretical results with experimental demonstration, that standard algorithms that are and are not rotationally invariant differ in their performance on real data, when the optimal solution is moderately sparse. Here we report experiments on the Fashion MNIST (Xiao et al., 2017) dataset. We use a multilayer feedforward network with two hidden layers of size 256 each, with two cases for the input layer weights: 1) fully connected (each 1st layer hidden node is connected to all inputs) and 2) "spindly" (each 1st layer hidden node is connected to all inputs via a copy of the network of Figure 1). On the unmodified dataset, both variants of the network achieve the same top-one accuracy of 85% (although the spindly version takes longer to train).

Next, we augment the data in two ways to create separate classes of features that differ in their informativeness. This generalizes our toy setup where exactly one feature is informative. First, we double the number of features of the examples by augmenting each example with uniformly sampled noise pixels in the range $[-1, 1]$. With noisy augmentation, the spindly network still achieves 85% test accuracy while the fully-connected network gets only to 71% (after a much longer training phase). In addition, the weights learned by the two networks are significantly different: The fully-connected network assigns almost the same magnitude of weights to the noisy features as to the image features, while the spindly network allocates much larger weights to the image features. This illustrates that rotation invariant algorithms have a harder time ignoring the noisy features.

Second, we augment each example on top of the noise with a one-hot representation of the 10 target class labels. This splits the features into three categories in terms of their informativeness: 1) highly informative label features, 2) less informative image features, and 3) noise features with no (structured) information. The spindly network achieves 100% test accuracy while the fully-connected network gets to 98%. We observe that the spindly network assigns much larger weights (in magnitude) to the label features while almost ignoring the rest. This phenomenon is less prominent in the fully-connected network. We defer all the details to Appendix G.

## 5. Open problems

An immediate question is whether the sparse classification problems that are provably hard to solve for rotation invariant neural net algorithms but are easy to solve for convolutional neural nets can be lifted to the regression setting (Li et al., 2021). Note that the existing lower bounds are for the less interesting case where the number of examples is less than the VC dimension. The question is whether with the addition of noise, there are still gaps towards the optimal algorithm when the number of examples is larger than the VC dimension.

Another interesting question is when transforming the instances (i.e. a kernelization) "helps" rotation invariant algorithms. Note that linear neurons that are fed transformed instances when updated with gradient descent again have the $1 - k/d$ lower bound in the noise-free under-constraint case (Warmuth and Vishwanathan, 2005). Proving similar lower bounds for the noisy over-constraint case as done in this paper would be a first step.

Also, the upper bounds proven in this paper focus (with the exception of priming) on mirror descent and its approximations. However there is a large body of work based on $L_1$ regularization that is also biased towards sparsity (See e.g. Tibshirani (2015); Axiotis and Yasuda (2023); Hoff (2017)). How these updates are related to mirror descent is an interesting research topic.

Recall that multiplicative updates are mirror descent based on the $\log$ link function. By stretching the $\mathrm{arcsinh}$ link for the two sided version of multiplicative updates (a.k.a. the EG$^\pm$ update (Warmuth, 2007)), Ghai et al. (2020) showed that this update has gradient descent as a special case. Actually we

observe that stretching (and shifting) can be used to enhance any mirror descent update with hyper parameters that realize gradient descent as a special case. The reason is that stretching turns any link into a linear link, which is the link function for gradient descent. However no practical algorithm have been developed based on this simple observation.

The "full versions" of many common optimization algorithms (Abdulkadirov et al., 2023) such as AdaGrad, NGD, Adam, RMS Prop are all rotation invariant. However in practice diagonalized versions of these updates are used. In a major step forward, we were able to solve the differential equations for all these algorithms, when $\boldsymbol{X}^\top \boldsymbol{X} = \lambda \boldsymbol{I}$ (the cases of arbitrary input covariance are challenging open problems). This lets us analyze the inductive biases of all these algorithms on linear neurons based on their gradient flow trajectories. For example, the diagonalized AdaGrad is biased away from sparsity. So using sparsity as a yard stick, we show that already for linear neurons, dramatic differences can occur compared to the optimal algorithms. The question is whether these differences remain when running the same algorithms on neural nets with any number of hidden layers.

We gave a lower bound for rotation invariant neural nets for learning noisy sparse linear problems. Our work suggests the following approach: The structure of the network plus the weight update imply a certain set of invariances and the invariances lead to lower bounds for the Bayes optimal algorithm of the model constructed from the invariances. For example, a certain two-sided invariance characterizes the matrix version of multiplicative updates (Warmuth et al., 2014). Is it thus possible to prove lower bounds for these algorithms for dense linear targets that are beaten by vanilla gradient descent?

We proved novel upper bounds on the excess risk of several non-rotation invariant algorithms. Notably, the upper bounds for the Approximated EGU$^\pm$ (Appendix D.2) and the spindly network (Appendix D.3) are of order $O(\sigma^2 \frac{\log d}{md})$, which is $O(\log d)$ worse than the corresponding bound for the priming method (Appendix D.4). It remains an open question whether this additional $O(\log d)$ factor is intrinsic to the algorithms or if it could be eliminated by a tighter error analysis for Approximated EGU$^\pm$ and spindly.

Transformers clearly have an ability to access individual tokens. The question is which structural feature of transformers enables them to do that and learn sparse linear problems. Partial results along this line recently appeared in Abernethy et al. (2023).

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

## Appendix A. Proof of the lower bound Theorem 1

**Proof** For any orthogonal $\boldsymbol{U}$, the algorithm's expected loss is

$$
\begin{aligned}
L_{\widehat{y}}(q_{\boldsymbol{U}}) &= \mathbb{E}_{\tilde{\boldsymbol{X}} \sim p_{\mathrm{in}}, \tilde{\boldsymbol{y}} \sim q_{\boldsymbol{U}}(\cdot|\tilde{\boldsymbol{X}}), Z}[\mathcal{L}(\widehat{y}(\boldsymbol{x}_{\mathrm{te}}|\boldsymbol{X}, \boldsymbol{y}), y_{\mathrm{te}})] \\
&= \mathbb{E}_{\tilde{\boldsymbol{X}} \sim p_{\mathrm{in}}, \tilde{\boldsymbol{y}} \sim q(\cdot|\tilde{\boldsymbol{X}}\boldsymbol{U}^\top), Z}[\mathcal{L}(\widehat{y}(\boldsymbol{x}_{\mathrm{te}}|\boldsymbol{X}, \boldsymbol{y}), y_{\mathrm{te}})] \\
&= \mathbb{E}_{\tilde{\boldsymbol{X}}' \sim p_{\mathrm{in}}, \tilde{\boldsymbol{y}} \sim q(\cdot|\tilde{\boldsymbol{X}}'), Z}[\mathcal{L}(\widehat{y}(\boldsymbol{x}'_{\mathrm{te}}|\boldsymbol{X}', \boldsymbol{y}), y_{\mathrm{te}})] \\
&= L_{\widehat{y}}(q),
\end{aligned}
$$

where $\tilde{\boldsymbol{X}}' = \tilde{\boldsymbol{X}}\boldsymbol{U}^\top$ (we also used $\boldsymbol{X}' = \boldsymbol{X}\boldsymbol{U}^\top$ and $\boldsymbol{x}'_{\mathrm{te}} = \boldsymbol{U}\boldsymbol{x}_{\mathrm{te}}$), and where the fourth line uses rotational symmetry of $p_{in}$ and rotational invariance of $\widehat{y}$. Therefore, presented with the problem $\mathring{q}$, the algorithm will achieve expected loss $L_{\widehat{y}}(q)$ regardless of the value of $\boldsymbol{U}$. This implies that $L_{\widehat{y}}(q)$ cannot be less than the optimal value $L_{\mathcal{B}}(\mathring{q})$. $\blacksquare$

## Appendix B. Proof of lower bound Theorem 2

Before we give the proof observe that one can equivalently think of the $m$ stacked copies of $\boldsymbol{H}$ as just a single copy of $\boldsymbol{H}$ with the variance of label noise reduced by a factor of $m$. Indeed, if $\boldsymbol{y}_i$ represents the label vector of size $d$ associated with the $i$-th copy of $\boldsymbol{H}$, then the training loss for any weight vector $\widehat{\boldsymbol{w}}$ is effectively

$$
\|\boldsymbol{y} - \boldsymbol{X}\widehat{\boldsymbol{w}}\|^2 = \sum_{i=1}^{m} \|\boldsymbol{y}_i - \boldsymbol{H}\widehat{\boldsymbol{w}}\|^2 = m\|\bar{\boldsymbol{y}} - \boldsymbol{H}\widehat{\boldsymbol{w}}\|^2 + \sum_{i=1}^{m} \|\boldsymbol{y}_i - \bar{\boldsymbol{y}}\|^2,
$$

where $\bar{\boldsymbol{y}} = m^{-1}\sum_i \boldsymbol{y}_i \sim \mathrm{N}(\boldsymbol{H}\boldsymbol{e}_1, \sigma^2/m\boldsymbol{I}_d)$ and the last term on the right-hand side does not depend on $\widehat{\boldsymbol{w}}$. Therefore, by varying $m$, we effectively alter the level of noise the algorithms encounter. We also note that our model is equivalent to the Gaussian Sequence Model (see, for example, Rigollet and Hütter (2023)), given by $\boldsymbol{y} = \boldsymbol{\theta} + \boldsymbol{\xi}$ (with $\boldsymbol{\theta} = \boldsymbol{X}\boldsymbol{e}_1$ and $\boldsymbol{\xi} \sim \mathcal{N}(\boldsymbol{0}, \sigma^2/m\boldsymbol{I}_d)$), a commonly used model for analyzing nonparametric and high dimensional statistical problems.

**Proof** We consider the rotated observational model described in the main text, which can be defined as follows. Let $\boldsymbol{w} \in \mathbb{R}^d$ be a weight vector drawn uniformly from a unit sphere $\mathcal{S}^{d-1} = \{\boldsymbol{w} \in \mathbb{R}^d : \|\boldsymbol{w}\| = 1\}$. The algorithm is given data set $(\boldsymbol{X}, \boldsymbol{y})$ with $\boldsymbol{X} = [\boldsymbol{H}; \ldots; \boldsymbol{H}]$ being $m$ copies of $\boldsymbol{H} = \sqrt{d}\boldsymbol{V}$, and $\boldsymbol{y} = \boldsymbol{H}\boldsymbol{w} + \boldsymbol{\xi}$, where $\boldsymbol{\xi} \sim \mathrm{N}(\boldsymbol{0}, \sigma^2\boldsymbol{I}_{dm})$ is a vector of Gaussian i.i.d. noise variables, each having zero mean and variance $\sigma^2$. Given $\boldsymbol{y}$, the algorithm is supposed to produce a vector of predictions $\widehat{\boldsymbol{y}} \in \mathbb{R}^d$ and is evaluated by means of the squared error, $e(\widehat{\boldsymbol{y}}|\boldsymbol{w}) = \frac{1}{d}\|\widehat{\boldsymbol{y}} - \boldsymbol{H}\boldsymbol{w}\|^2$. We first note that without loss of generality, the algorithm produces a weight vector $\widehat{\boldsymbol{w}}$, based on which the predictions are generated, $\widehat{\boldsymbol{y}} = \boldsymbol{H}\widehat{\boldsymbol{w}}$; this is due to the fact that $\boldsymbol{H}$ is invertible (as multiplicity of an orthogonal matrix), so for every $\widehat{\boldsymbol{y}}$, one can have a corresponding weight vector $\widehat{\boldsymbol{w}} = \boldsymbol{H}^{-1}\widehat{\boldsymbol{y}}$. Thus, using (3) the error can be equivalently written as

$$
e(\widehat{\boldsymbol{w}}|\boldsymbol{w}) = \|\widehat{\boldsymbol{w}} - \boldsymbol{w}\|^2.
$$

It is well-known (see, e.g., Berger (1985)) that the expected squared error, $\mathbb{E}_{\boldsymbol{w}, \boldsymbol{\xi}}[e(\widehat{\boldsymbol{w}}|\boldsymbol{w})]$ (with expectation with respect to the prior and the label noise) is minimized by the *posterior mean*

$\widehat{\boldsymbol{w}}^{\star} = \mathbb{E}_{\boldsymbol{w}|\boldsymbol{y}}[\boldsymbol{w}]$, that is the mean value of $\boldsymbol{w}$ with respect to the posterior distribution $q(\boldsymbol{w}|\boldsymbol{X}, \boldsymbol{y})$. Even though the posterior mean does not have a nice analytic form, we can still lower bound its expected squared error using a technique borrowed from (Marchand, 1993; Dicker, 2016). Let $\widehat{\boldsymbol{w}}_{RR}$ be the ridge regression estimator:

$$\widehat{\boldsymbol{w}}_{RR} = (\boldsymbol{X}^{\top}\boldsymbol{X} + \sigma^2 d\boldsymbol{I})^{-1}\boldsymbol{X}^{\top}\boldsymbol{y},$$

which is the posterior mean itself (and thus optimal) when the prior over $\boldsymbol{w}$ is Gaussian with zero mean and covariance $\frac{1}{d}\boldsymbol{I}$ (the covariance is multiplied by factor $d^{-1}$ to have $\mathbb{E}[\|\boldsymbol{w}\|^2] = \mathbb{E}[\boldsymbol{w}\boldsymbol{w}^{\top}] = \text{tr}(\frac{1}{d}\boldsymbol{I}) = 1$ as in the unit sphere prior case). Even though the Gaussian prior differs from the uniform prior over a unit sphere, it turns out that the RR predictor has error only slightly larger than that of the optimal Bayes predictor $\widehat{\boldsymbol{w}}^{\star}$:

**Lemma 4**

$$\mathbb{E}_{\boldsymbol{w},\boldsymbol{\xi}}\left[e(\widehat{\boldsymbol{w}}_{RR}|\boldsymbol{w})\right] \leq \mathbb{E}_{\boldsymbol{w},\boldsymbol{\xi}}\left[e(\widehat{\boldsymbol{w}}^{\star}|\boldsymbol{w})\right] + \frac{1}{d}\frac{\sigma^2}{\sigma^2 + m}.$$

**Proof** Dicker (2016) considered a Bayesian setting similar to ours, with $\sigma^2 = 1$ and $\boldsymbol{w}$ distributed uniformly over $\tau\mathcal{S}^{d-1} = \{\boldsymbol{w} : \|\boldsymbol{w}\| = \tau\}$. To account for this setting, we note that in our setup,

$$\sigma^{-1}\boldsymbol{y} = \boldsymbol{X}^{\top}(\sigma^{-1}\boldsymbol{w}) + \underbrace{\sigma^{-1}\boldsymbol{\xi}}_{\sim N(\boldsymbol{0},\boldsymbol{I}_{dm})},$$

so that we can set $\tau = \sigma^{-1}$ and assume unit variance of the noise. Their 'oracle ridge estimator' is thus $\widehat{\boldsymbol{w}}_{RR}$. Furthermore, since $\|\widehat{\boldsymbol{w}} - \boldsymbol{w}\|^2 = \sigma^2\|\sigma^{-1}\widehat{\boldsymbol{w}} - \sigma^{-1}\boldsymbol{w}\|^2$, we need to multiply their bound by $\sigma^2$. We use their Theorem 2 (adapted to the modifications stated above):

**Theorem 5** *(Theorem 2 by Dicker (2016), Theorem 3.1 by Marchand (1993))* Let $n = md$ and let $s_1 \geq \ldots \geq s_d$ denote the eigenvalues of $n^{-1}\boldsymbol{X}^{\top}\boldsymbol{X}$. Then

$$\mathbb{E}_{\boldsymbol{w},\boldsymbol{\xi}}\left[e(\widehat{\boldsymbol{w}}_{RR}|\boldsymbol{w})\right] \leq \mathbb{E}_{\boldsymbol{w},\boldsymbol{\xi}}\left[e(\widehat{\boldsymbol{w}}^{\star}|\boldsymbol{w})\right] + \frac{\sigma^2}{d}\frac{s_1}{s_d}\text{tr}\left\{(\boldsymbol{X}^{\top}\boldsymbol{X} + d\sigma^2\boldsymbol{I}_n)^{-1}\right\}.$$

Since $\boldsymbol{X}^{\top}\boldsymbol{X} = n\boldsymbol{I}$, we have $s_1 = s_d = 1$ and $(\boldsymbol{X}^{\top}\boldsymbol{X} + \boldsymbol{I}_n)^{-1} = (n + d\sigma^2)^{-1}\boldsymbol{I}_n$, and thus,

$$\mathbb{E}_{\boldsymbol{w},\boldsymbol{\xi}}\left[e(\widehat{\boldsymbol{w}}_{RR}|\boldsymbol{w})\right] \leq \mathbb{E}_{\boldsymbol{w},\boldsymbol{\xi}}\left[e(\widehat{\boldsymbol{w}}^{\star}|\boldsymbol{w})\right] + \frac{\sigma^2}{n + d\sigma^2} = \mathbb{E}_{\boldsymbol{w},\boldsymbol{\xi}}\left[e(\widehat{\boldsymbol{w}}^{\star}|\boldsymbol{w})\right] + \frac{1}{d}\frac{\sigma^2}{m + \sigma^2}. \qquad \blacksquare$$

Now, we compute the expected error of $\widehat{\boldsymbol{w}}_{RR}$. Since $\boldsymbol{X}^{\top}\boldsymbol{X} = md\boldsymbol{I}$, we get

$$\widehat{\boldsymbol{w}}_{RR} = \frac{1}{md + \sigma^2 d}\boldsymbol{X}^{\top}\boldsymbol{y} = \frac{1}{md + \sigma^2 d}\boldsymbol{X}^{\top}(\boldsymbol{X}\boldsymbol{w} + \boldsymbol{\xi}) = \frac{md\boldsymbol{w} + \boldsymbol{X}^{\top}\boldsymbol{\xi}}{md + \sigma^2 d},$$

and thus

$$e(\widehat{\boldsymbol{w}}_{RR}|\boldsymbol{w}) = \|\widehat{\boldsymbol{w}}_{RR} - \boldsymbol{w}\|^2 = \left\|\frac{\boldsymbol{X}^{\top}\boldsymbol{\xi} - \sigma^2 d\boldsymbol{w}}{md + \sigma^2 d}\right\|^2$$

$$= \frac{\|\boldsymbol{X}^{\top}\boldsymbol{\xi}\|^2}{(md + \sigma^2 d)^2} - \frac{md\boldsymbol{w}^{\top}\boldsymbol{X}^{\top}\boldsymbol{\xi}}{(md + \sigma^2 d)^2} + \frac{\sigma^4 d^2\|\boldsymbol{w}\|^2}{(md + \sigma^2 d)^2}.$$

We take the expectation over $\boldsymbol{w}$, under which the middle term in the last line vanishes (as $\mathbb{E}[\boldsymbol{w}] = \boldsymbol{0}$ over a unit sphere) and use $\|\boldsymbol{w}\| = 1$ to get

$$\mathbb{E}_{\boldsymbol{w}}[e(\widehat{\boldsymbol{w}}_{RR}|\boldsymbol{w})] = \frac{\|\boldsymbol{X}^\top \boldsymbol{\xi}\|^2}{(md + \sigma^2 d)^2} + \frac{\sigma^4 d^2}{(md + \sigma^2 d)^2}.$$

We further take an expectation over $\boldsymbol{\xi}$ and use

$$\mathbb{E}[\|\boldsymbol{X}^\top \boldsymbol{\xi}\|^2] = \mathbb{E}[\operatorname{tr}(\boldsymbol{X}^\top \boldsymbol{\xi}\boldsymbol{\xi}^\top \boldsymbol{X})] = \operatorname{tr}(\boldsymbol{X}^\top \mathbb{E}[\boldsymbol{\xi}\boldsymbol{\xi}^\top]\boldsymbol{X}) = \operatorname{tr}(\boldsymbol{X}^\top \boldsymbol{X}) = md \operatorname{tr}(\boldsymbol{I}) = md^2,$$

to get:

$$\mathbb{E}_{\boldsymbol{w},\boldsymbol{\xi}}[e(\widehat{\boldsymbol{w}}_{RR}|\boldsymbol{w})] = \frac{md^2}{(md + \sigma^2 d)^2} + \frac{\sigma^4 d^2}{(md + \sigma^2 d)^2}. = \frac{\sigma^2 d(md + \sigma^2 d)}{(md + \sigma^2 d)^2} = \frac{\sigma^2}{m + \sigma^2}.$$

Using this together with Lemma 4 gives the lower bound on the Bayes optimal predictor in the rotated observational model.

$$\mathbb{E}_{\boldsymbol{w},\boldsymbol{\xi}}[e(\widehat{\boldsymbol{w}}^\star|\boldsymbol{w})] \geq \frac{d-1}{d} \frac{\sigma^2}{\sigma^2 + m}.$$

We now use the fact that that $\boldsymbol{H} = \sqrt{d}\boldsymbol{V}$ with orthogonal matrix $\boldsymbol{V}$ of size $d \times d$, drawn uniformly at random (with respect to Haar measure), so that our input distribution is rotation symmetric. This means that we can apply Theorem 1 and conclude that any rotation invariant algorithm has the expected error at least $\frac{d-1}{d} \frac{\sigma^2}{\sigma^2 + m}$ on the original problem, that is for $\boldsymbol{w} = \boldsymbol{e}_1$. ∎

Note that the proof would significantly simplify if we assumed from the start that the target weight vector $\boldsymbol{w}$ is generated from a Gaussian distribution $N(\boldsymbol{0}, \frac{1}{d}\boldsymbol{I}_d)$ rather than from a unit sphere $\mathcal{S}^{d-1}$ (both priors give unit squared norm of $\boldsymbol{w}$ on expectation), as the Bayes predictor would be exactly the RR predictor, giving even a better lower bound of $\frac{\sigma^2}{\sigma^2 + m}$, without the need to apply results from Marchand (1993); Dicker (2016). This would, however, result in a random norm of the sparse target vector. In our proof we opted for a bound with a fixed, unit norm of $\boldsymbol{w}$.

## Appendix C. Lower bound for i.i.d. sampling

Instead of observing copies of the same matrix $\boldsymbol{H}$, one can instead consider a standard linear model framework, where individual inputs $\boldsymbol{x}$ are drawn i.i.d. from some input distribution $p_{\mathrm{in}}(\boldsymbol{x})$. The associated labels are given by $y = \boldsymbol{x}_i^\top \boldsymbol{e}_1 + \xi$, where $\xi \sim \mathcal{N}(0, \sigma^2)$. The algorithm observes $n$ such i.i.d. samples $(\boldsymbol{X}, \boldsymbol{y})$ and produces a weight vector $\widehat{\boldsymbol{w}}$, which is then evaluated by the squared error $\|\widehat{\boldsymbol{w}} - \boldsymbol{e}_1\|^2$. Assume $p_{\mathrm{in}}(\boldsymbol{x})$ is rotationally symmetric with covariance $\mathbb{E}[\boldsymbol{x}\boldsymbol{x}^\top] = \boldsymbol{I}_d$ (so that the expected input size matches that from our previous setup); for instance, we could have $\boldsymbol{x} \sim \mathcal{N}(\boldsymbol{0}, \boldsymbol{I}_d)$. Employing our Bayesian argument from section 2.2, one can establish a similar lower bound for any rotation-invariant algorithm. Note that for large $d$, the bound is essentially the same as that of Theorem 2. The proof makes use of information-theoretic arguments by Gerchinovitz et al. (2020).

**Theorem 6** *Let the rows of $\boldsymbol{X}$ be drawn i.i.d. from rotationally symmetric $p_{\mathrm{in}}$ with covariance $\boldsymbol{I}_d$, and the labels given by $\boldsymbol{y} = \boldsymbol{X}\boldsymbol{e}_1 + \boldsymbol{\xi}$, $\boldsymbol{\xi} \sim \mathcal{N}(\boldsymbol{0}, \sigma^2 \boldsymbol{I}_n)$. Assume $d \geq 12$. Then, the expected error (with a random choice of the sample) of any rotation-invariant learning algorithm is at least*

$$\mathbb{E}[e(\widehat{\boldsymbol{w}})] \geq \left(\frac{\sigma^2}{e(\sigma^2 + m/2)}\right)^{d/(d-1)}.$$

**Proof** We consider a Bayesian setup, in which the target weight vector is drawn uniformly from a unit sphere, $\boldsymbol{w} \sim \mathcal{S}^{d-1}$, with $\mathcal{S}^{d-1} = \{\boldsymbol{w} \in \mathbb{R}^d \colon \|\boldsymbol{w}\| = 1\}$. We will show a lower bound on the error of *any* predictor $\widehat{\boldsymbol{w}}$, $\mathbb{E}_{\boldsymbol{w},\boldsymbol{\xi}}\left[\|\widehat{\boldsymbol{w}} - \boldsymbol{w}\|^2\right]$, by exploiting the fact that it is not less than the error of the Bayes optimal predictor $\widehat{\boldsymbol{w}}^\star$ (the posterior mean), followed by a lower bound on the error of $\widehat{\boldsymbol{w}}^\star$:

$$\mathbb{E}_{\boldsymbol{w},\boldsymbol{\xi}}\left[\|\widehat{\boldsymbol{w}}^\star - \boldsymbol{w}\|^2\right] \geq \left(\frac{\sigma^2}{e(\sigma^2 + m/2)}\right)^{d/(d-1)}. \tag{8}$$

Using the rotational symmetry of the input distribution $p_{\text{in}}$, and giving the same arguments as in the proof of Theorem 2 (or Section 2.2), this will imply the same lower bound (8) for any rotation-invariant algorithm with a sparse target vector $\boldsymbol{w} = \boldsymbol{e}_1$.

Unfortunately, for prior uniform over a sphere, $\widehat{\boldsymbol{w}}^\star$ does not have a closed form. We will, however, use a continuous version of Fano's inequality (Gerchinovitz et al., 2020) to lower bound its Bayes risk.

From now on, we condition our analysis on $\boldsymbol{X}$ drawn i.i.d. from the input distribution. We lower-bound the Bayes risk using Markov's inequality: for any $\epsilon > 0$,

$$\mathbb{E}[\|\widehat{\boldsymbol{w}}^\star - \boldsymbol{w}\|^2] \geq \epsilon P(\|\widehat{\boldsymbol{w}}^\star - \boldsymbol{w}\|^2 \geq \epsilon) = \epsilon \left(1 - P(\|\widehat{\boldsymbol{w}}^\star - \boldsymbol{w}\|^2 < \epsilon)\right). \tag{9}$$

where the expectation and the probability is with respect to random $\boldsymbol{y}, \boldsymbol{w}$ (conditioned on $\boldsymbol{X}$). Let $P_{\boldsymbol{w}} = \mathcal{N}(\boldsymbol{X}\boldsymbol{w}, \sigma^2\boldsymbol{I}_n)$ be the distribution of $\boldsymbol{y}$ given $\boldsymbol{w}$ (and $\boldsymbol{X}$). Given $\boldsymbol{w}$, define event $\mathcal{A}_{\boldsymbol{w}} = \{\|\widehat{\boldsymbol{w}}^\star - \boldsymbol{w}\|^2 < \epsilon\}$. We can write

$$P\left(\|\widehat{\boldsymbol{w}}^\star - \boldsymbol{w}\|^2 < \epsilon\right) = \mathbb{E}_{\boldsymbol{w}}\left[P_{\boldsymbol{w}}(\mathcal{A}_{\boldsymbol{w}})\right]$$

Using techniques from Gerchinovitz et al. (2020), we have for any distribution $Q$ over $\boldsymbol{y}$,

$$\mathbb{E}_{\boldsymbol{w}}\left[P_{\boldsymbol{w}}(\mathcal{A}_{\boldsymbol{w}})\right] \leq \mathbb{E}_{\boldsymbol{w}}\left[Q(\mathcal{A}_{\boldsymbol{w}})\right] + \sqrt{\left(\mathbb{E}_{\boldsymbol{w}}\left[Q(\mathcal{A}_{\boldsymbol{w}})\right]\right)\left(\mathbb{E}_{\boldsymbol{w}}\left[\chi^2(P_{\boldsymbol{w}}, Q)\right]\right)} \tag{10}$$

where $\chi^2(P, Q) = \mathbb{E}_Q\left[\left(\frac{dP}{dQ}\right)^2\right]$ is the $\chi^2$-divergence between $P$ and $Q$. We now choose $Q = \mathcal{N}(\boldsymbol{0}, \boldsymbol{\Sigma})$ with $\boldsymbol{\Sigma} = \sigma^2\boldsymbol{I}_n + \frac{1}{d}\boldsymbol{X}\boldsymbol{X}^\top$. It follow from the Gaussian integration that

$$\chi^2(P_{\boldsymbol{w}}, Q) = \sqrt{\det\left(\sigma^{-2}\boldsymbol{\Sigma}^2(2\boldsymbol{\Sigma} - \sigma^2\boldsymbol{I}_n)^{-1}\right)} e^{\boldsymbol{w}^\top\boldsymbol{X}^\top\left(2\boldsymbol{\Sigma} - \sigma^2\boldsymbol{I}_n\right)^{-1}\boldsymbol{X}\boldsymbol{w}} - 1.$$

First note that

$$\sigma^{-2}\boldsymbol{\Sigma}^2(2\boldsymbol{\Sigma} - \sigma^2\boldsymbol{I}_n)^{-1} = \left(\boldsymbol{I}_n + \frac{1}{d\sigma^2}\boldsymbol{X}\boldsymbol{X}^\top\right)^2 \left(\boldsymbol{I}_n + \frac{2}{d\sigma^2}\boldsymbol{X}\boldsymbol{X}^\top\right)^{-1} \preceq \boldsymbol{I}_n + \frac{1}{2d\sigma^2}\boldsymbol{X}\boldsymbol{X}^\top.$$

Indeed, by taking any eigenvalue $\lambda_i$ of $\frac{1}{2d\sigma^2}\boldsymbol{X}\boldsymbol{X}^\top$,

$$\frac{(1 + \lambda_i)^2}{1 + 2\lambda_i} \leq 1 + \frac{\lambda_i^2}{1 + 2\lambda_i} \leq 1 + \frac{\lambda_i}{2}.$$

Moreover, from the property of the determinant, that $\det(\boldsymbol{I}_m + \boldsymbol{A}\boldsymbol{B}) = \det(\boldsymbol{I}_n + \boldsymbol{B}\boldsymbol{A})$ for $\boldsymbol{A}$ of size $m \times n$ and $\boldsymbol{B}$ of size $n \times m$, we get

$$\det\left(\boldsymbol{I}_n + \frac{1}{2d\sigma^2}\boldsymbol{X}\boldsymbol{X}^\top\right) = \det\left(\boldsymbol{I}_m + \frac{1}{2d\sigma^2}\boldsymbol{X}^\top\boldsymbol{X}\right).$$

Finally, note that

$$\boldsymbol{X}^\top(2\boldsymbol{\Sigma} - \sigma^2\boldsymbol{I}_n)^{-1}\boldsymbol{X} = \boldsymbol{X}^\top\left(\sigma^2\boldsymbol{I}_n + \frac{2}{d}\boldsymbol{X}\boldsymbol{X}^\top\right)^{-1}\boldsymbol{X} \preceq \frac{d}{2}\boldsymbol{X}^\top(\boldsymbol{X}\boldsymbol{X}^\top)^\dagger\boldsymbol{X} = \frac{d}{2}\boldsymbol{I}_d,$$

so that

$$e^{\boldsymbol{w}^\top\boldsymbol{X}^\top(2\boldsymbol{\Sigma} - \sigma^2\boldsymbol{I}_n)^{-1}\boldsymbol{X}\boldsymbol{w}} \le e^{\frac{d}{2}\|\boldsymbol{w}\|^2} = e^{\frac{d}{2}}.$$

Thus, we get:

$$\chi^2(P_{\boldsymbol{w}}, Q) \le \sqrt{\det\left(\boldsymbol{I}_m + \frac{1}{2d\sigma^2}\boldsymbol{X}^\top\boldsymbol{X}\right)}e^{\frac{d}{2}}$$

For any positive-definite $d \times d$ matrix $\boldsymbol{A}$ with eigenvalues $a_1, \ldots, a_d$, we have $\ln\det\boldsymbol{A} = \sum_i \ln a_i$. Using the concavity of negative logarithm, we can bound:

$$\ln\det\boldsymbol{A} = d\frac{1}{d}\sum_i \ln(a_i) \le d\ln\left(\frac{1}{d}\sum_i a_i\right) = d\ln\operatorname{tr}(\boldsymbol{A}/d),$$

which gives $\det\boldsymbol{A} \le \operatorname{tr}(\boldsymbol{A}/d)^d$ and thus

$$\chi^2(P_{\boldsymbol{w}}, Q) \le \left(e\operatorname{tr}\left(\frac{1}{d}\boldsymbol{I}_m + \frac{1}{2d^2\sigma^2}\boldsymbol{X}^\top\boldsymbol{X}\right)\right)^{\frac{d}{2}}.$$

Since this expression does not depend on $\boldsymbol{w}$, taking expectation on both sides (with respect to $\boldsymbol{w}$) gives the same bound on $\mathbb{E}_{\boldsymbol{w}}\left[\chi^2(P_{\boldsymbol{w}}, Q)\right]$.

We will now upper-bound $\mathbb{E}_{\boldsymbol{w}}[Q(\mathcal{A}_{\boldsymbol{w}})]$. Let $\mathbf{1}\{C\}$ denote the Iverson bracket which is 1 if $C$ holds and 0 otherwise. Using the fact that $\widehat{\boldsymbol{w}}^\star = \widehat{\boldsymbol{w}}^\star(\boldsymbol{y})$ depends on $\boldsymbol{y}$ (and $\boldsymbol{X}$), but not on $\boldsymbol{w}$, and that $\boldsymbol{y} \sim Q$ is independent of $\boldsymbol{w}$, we have

$$\mathbb{E}_{\boldsymbol{w}}[Q(\mathcal{A}_{\boldsymbol{w}})] = \mathbb{E}_{\boldsymbol{w}}[Q(\|\widehat{\boldsymbol{w}}^\star - \boldsymbol{w}\|^2 < \epsilon)] = \mathbb{E}_{\boldsymbol{w}\sim\mathcal{S}^{d-1}, \boldsymbol{y}\sim Q}\left[\mathbf{1}\left\{\|\widehat{\boldsymbol{w}}^\star(\boldsymbol{y}) - \boldsymbol{w}\|^2 < \epsilon\right\}\right]$$

$$= \mathbb{E}_{\boldsymbol{y}\sim Q}\left[\mathbb{E}_{\boldsymbol{w}\sim\mathcal{S}^{d-1}}\left[\mathbf{1}\left\{\|\widehat{\boldsymbol{w}}^\star(\boldsymbol{y}) - \boldsymbol{w}\|^2 < \epsilon\right\}\right]\right]$$

$$\le \mathbb{E}_{\boldsymbol{y}\sim Q}\left[\max_{\widehat{\boldsymbol{w}}}\mathbb{E}_{\boldsymbol{w}\sim\mathcal{S}^{d-1}}\left[\mathbf{1}\left\{\|\widehat{\boldsymbol{w}} - \boldsymbol{w}\|^2 < \epsilon\right\}\right]\right] = \max_{\widehat{\boldsymbol{w}}}\mathbb{E}_{\boldsymbol{w}}\left[\mathbf{1}\left\{\|\widehat{\boldsymbol{w}} - \boldsymbol{w}\|^2 < \epsilon\right\}\right],$$

where in the last equality we used the fact that the term inside the expectation over $\boldsymbol{y}$ does not depend on $\boldsymbol{y}$. Let us now fix $\widehat{\boldsymbol{w}}$ and we will bound $\mathbb{E}_{\boldsymbol{w}}\left[\mathbf{1}\left\{\|\widehat{\boldsymbol{w}} - \boldsymbol{w}\|^2 < \epsilon\right\}\right]$ Since the distribution of $\boldsymbol{w}$ is rotation-invariant (uniform over a sphere), without loss of generality take $\widehat{\boldsymbol{w}} = c\boldsymbol{e}_1$ with $c > 0$ ($c = 0$ would give 0 for any $\epsilon < 1$). We have

$$\|\widehat{\boldsymbol{w}}^\star - \boldsymbol{w}\|^2 = (c - w_1)^2 + \sum_{j=2}^d w_j^2 = (c - w_1)^2 + 1 - w_1^2 = 1 + c^2 - 2cw_1,$$

and thus

$$\mathbb{E}_{\boldsymbol{w}}\left[\mathbf{1}\left\{\|\widehat{\boldsymbol{w}} - \boldsymbol{w}\|^2 < \epsilon\right\}\right] = \mathbb{E}_{\boldsymbol{w}}\left[\mathbf{1}\left\{1 + c^2 - 2cw_1 < \epsilon\right\}\right] = P\left(w_1 > \frac{1 + c^2 - \epsilon}{2c}\right).$$

Since the probability is nonincreasing in $\frac{1+c^2-\epsilon}{2c}$ we take $c$ which minimizes this quantity to get an upper bound. By inspecting the derivative

$$\min_c \frac{1+c^2-\epsilon}{2c} = \frac{1}{2}\min_c c + \frac{1-\epsilon}{c} \overset{c=\sqrt{1-\epsilon}}{=} \sqrt{1-\epsilon},$$

we have

$$\mathbb{E}_{\boldsymbol{w}}\left[\mathbf{1}\left\{\|\widehat{\boldsymbol{w}}-\boldsymbol{w}\|^2 < \epsilon\right\}\right] = P\left(w_1 > \sqrt{1-\epsilon}\right) = \frac{1}{2}P\left(w_1^2 > 1-\epsilon\right),$$

where we used the fact that distribution of $w_1$ is symmetric around zero. To determine the distribution of $w_1^2$ we use the fact that drawing $\boldsymbol{w}$ uniformly over a unit sphere amounts to drawing $\boldsymbol{u} \sim \mathcal{N}(\boldsymbol{0}, \boldsymbol{I}_d)$ and setting $\boldsymbol{w} = \frac{\boldsymbol{u}}{\|\boldsymbol{u}\|}$. Thus, $w_1^2 = \frac{x}{x+y}$, where $x \sim \chi^2(1) = \mathrm{Gamma}(\alpha = 1/2, \beta = 1/2)$ and $y \sim \chi^2(d-1) = \mathrm{Gamma}(\alpha = (d-1)/2, \beta = 1/2)$ and $x, y$ independent, Thus, $w_1^2$ is distributed according to beta distribution $B(\alpha = 1/2, \beta = (d-1)/2)$. This gives

$$P\left(w_1^2 > 1-\epsilon\right) = \frac{\Gamma(d/2)}{\Gamma(1/2)\Gamma((d-1)/2)}\int_{1-\epsilon}^1 x^{-1/2}(1-x)^{-(d-1)/2}\mathrm{d}x$$

$$\leq \frac{\Gamma(d/2)}{\Gamma(1/2)\Gamma((d-1)/2)}(1-\epsilon)^{-1/2}\int_{1-\epsilon}^1 (1-x)^{(d-1)/2-1}\mathrm{d}x$$

$$= \frac{\Gamma(d/2)}{\Gamma(1/2)\Gamma((d-1)/2)}\frac{2(1-\epsilon)^{-1/2}\epsilon^{(d-1)/2}}{d-1}$$

Assuming $\epsilon < 1/2$, we have $2(1-\epsilon)^{-1/2} \leq 2\sqrt{2}$. We also use Gautschi's inequality which states that for any $x > 0$ and any $s \in (0,1)$, $\frac{\Gamma(x+1)}{\Gamma(x+s)} \leq (x+1)^{1-s}$. Taking $x = \frac{d}{2}-1$ and $s = \frac{1}{2}$, we can bound $\frac{\Gamma(d/2)}{\Gamma((d-1)/2)} \leq \sqrt{d/2} \leq \sqrt{d-1}$, where we used $d \geq 2$. This, together with $\Gamma(1/2) = \sqrt{\pi}$ allows us to bound

$$P\left(w_1^2 > 1-\epsilon\right) \leq \frac{2\sqrt{2}\epsilon^{(d-1)/2}}{\sqrt{\pi}\sqrt{d-1}}.$$

Thus, $\mathbb{E}_{\boldsymbol{w}}[Q(\mathcal{A}_{\boldsymbol{w}})]$ can be bounded by

$$\mathbb{E}_{\boldsymbol{w}}[Q(\mathcal{A}_{\boldsymbol{w}})] = \frac{1}{2}P\left(w_1^2 > 1-\epsilon\right) \leq \frac{\sqrt{2}\epsilon^{(d-1)/2}}{\sqrt{\pi}\sqrt{d-1}}$$

Plugging all bounds to (10) gives:

$$\mathbb{E}_{\boldsymbol{w}}\left[P_{\boldsymbol{w}}(\mathcal{A}_{\boldsymbol{w}})\right] \leq \frac{\sqrt{2}\epsilon^{(d-1)/2}}{\sqrt{\pi(d-1)}} + \sqrt{\sqrt{2}\frac{\epsilon^{(d-1)/2}}{\sqrt{(\pi(d-1))}}\left(e\,\mathrm{tr}\left(\frac{1}{d}\boldsymbol{I}_m + \frac{1}{2d^2\sigma^2}\boldsymbol{X}^\top\boldsymbol{X}\right)\right)^{\frac{d}{2}}}.$$

Now, take

$$\epsilon^{-1} = \left(e\,\mathrm{tr}\left(\frac{1}{d}\boldsymbol{I}_m + \frac{1}{2d^2\sigma^2}\boldsymbol{X}^\top\boldsymbol{X}\right)\right)^{d/(d-1)}.$$

Note that $\epsilon \leq 1/e$. This gives

$$\mathbb{E}_{\boldsymbol{w}}\left[P_{\boldsymbol{w}}(\mathcal{A}_{\boldsymbol{w}})\right] \leq \frac{\sqrt{2}e^{-(d-1)/2}}{\sqrt{\pi(d-1)}} + \sqrt{\frac{\sqrt{2}}{\sqrt{\pi(d-1)}}} \leq \frac{1}{2},$$

whenever $d \geq 12$. We thus have shown

$$P\left(\|\widehat{\boldsymbol{w}}^{\star} - \boldsymbol{w}\|^2 \leq \epsilon\right) \leq \frac{1}{2},$$

which, using (9), gives:

$$\mathbb{E}_{\boldsymbol{w},\boldsymbol{y}|\boldsymbol{X}}[\|\widehat{\boldsymbol{w}}^{\star} - \boldsymbol{w}\|^2] \geq \frac{\epsilon}{2}.$$

To get a bound with respect to a random choice of $\boldsymbol{X}$, note that function $x \mapsto x^{-d/(d-1)}$ is convex, and thus:

$$\mathbb{E}[\|\widehat{\boldsymbol{w}}^{\star} - \boldsymbol{w}\|^2] \geq \frac{1}{2}\mathbb{E}[\epsilon] \geq \frac{1}{2}\left(e \operatorname{tr}\left(\frac{1}{d}\boldsymbol{I}_m + \frac{1}{2d^2\sigma^2}\mathbb{E}[\boldsymbol{X}^\top\boldsymbol{X}]\right)\right)^{d/(d-1)}$$

$$= \left(e\left(1 + \frac{n\tau^2}{2d\sigma^2}\right)\right)^{-d/(d-1)} = \left(\frac{\sigma^2}{e(\sigma^2 + n\tau^2/(2d))}\right)^{d/(d-1)}$$

∎

## Appendix D. Upper bounds

We now show how to break the above lower bound of $\frac{d-1}{d}\frac{\sigma^2}{\sigma^2+m}$ on the error (excess risk) of rotation invariant algorithms. We prove the upper bounds for four algorithms: EG$^{\pm}$ (Appendix D.1), Approximated EGU$^{\pm}$ (Appendix D.2), the spindly network (D.3), as well as the novel priming method (Warmuth and Amid, 2023) (Appendix D.4). We use the same setup as in the lower bound, i.e. $\boldsymbol{X}$ consists of $m$ stacked copies of a matrix $\boldsymbol{H} = \sqrt{d}\boldsymbol{V}$, when $\boldsymbol{V}$ is a rotation matrix and $\boldsymbol{y}$ is sparse linear (the first component) plus Gaussian noise with variance $\sigma^2$. The only difference is that for the lower bound, $\boldsymbol{V}$ was randomly chosen, but the upper bounds hold for *any* rotation matrix $\boldsymbol{V}$. The upper bound on the error that we obtain[13] is essentially $O(\sigma^2\frac{\log d}{md})$.

We then prove our main upper bound for the unnormalized EGU$^{\pm}$.

### D.1. Upper bound for the Exponentiated Gradient update

We begin with a bound for the normalized version of the multiplicative update algorithm. Specifically, we consider a batch version of the *2-sided Exponentiated Update algorithm* (EG$^{\pm}$) (Kivinen and Warmuth, 1997). The batch EG$^{\pm}$ algorithm maintains two vectors, $\boldsymbol{v}_t^+$ and $\boldsymbol{v}_t^-$, and its weight estimate is given by $\boldsymbol{w}_t = \boldsymbol{v}_t^+ - \boldsymbol{v}_t^-$. It starts with a set of weights $\boldsymbol{v}_1^+ = \boldsymbol{v}_1^- = \frac{1}{2d}\mathbf{1}$ (so that $\|\boldsymbol{v}_1^+\|_1 + \|\boldsymbol{v}_1^-\|_1 = 1$) and updates according to

$$\boldsymbol{v}_{t+1}^+ \propto \boldsymbol{v}_t^+ \odot e^{-\eta\nabla L(\boldsymbol{w}_t)}, \qquad \boldsymbol{v}_{t+1}^- \propto \boldsymbol{v}_t^- \odot e^{\eta\nabla L(\boldsymbol{w}_t)},$$

where $\odot$ is component-wise multiplication, and the normalization enforces $\|\boldsymbol{v}_{t+1}^+\|_1 + \|\boldsymbol{v}_{t+1}^-\|_1 = 1$, while $L(\boldsymbol{w})$ is the average total loss on the training sample:

$$L(\boldsymbol{w}) = \frac{1}{dm}\|\boldsymbol{X}\boldsymbol{w} - \boldsymbol{y}_t\|^2.$$

Here we prove that the batch version of EG$^{\pm}$ achieves small error already after one trial, when the learning rate is set to a sufficiently large value.

---

13. Note that at this point, we do not consider general input matrices $\boldsymbol{X}$ in the upper bounds. Allowing arbitrary covariance structure makes the analysis much more complicated and the general case is left for future research.

**Theorem 7** *The expected error of the batch* $EG^\pm$ *algorithm after the first iteration is bounded by*

$$e(\boldsymbol{w}_2) \leq 2de^{-\eta} + 8de^{-\frac{md}{32\sigma^2}}.$$

**Proof** The proof relies on the facts that (a) under large learning rate, the $EG^\pm$ weight estimate becomes argmax over the negative gradient coordinates, and (b) with high probability, the least component of the gradient is the first one, so that the argmax is $\boldsymbol{e}_1$. $EG^\pm$ makes use of the fact that the norm of the linear target $\boldsymbol{e}_1$ is 1 and this additional knowledge allows the speedup.[14]

We start with computing the gradient of the average loss:

$$\nabla L(\boldsymbol{w}) = \frac{2}{dm} \sum_{t=1}^m \sqrt{d} \boldsymbol{V}^\top (\sqrt{d} \boldsymbol{V} \boldsymbol{w} - \boldsymbol{y}_t) = \frac{2}{dm} \sum_{t=1}^m \sqrt{d} \boldsymbol{V}^\top (\sqrt{d} \boldsymbol{V}(\boldsymbol{w} - \boldsymbol{e}_1) - \boldsymbol{\xi}_t)$$

$$= \frac{2}{dm} \sum_{t=1}^m \left( d(\boldsymbol{w} - \boldsymbol{e}_1) + \sqrt{d} \boldsymbol{V}^\top \boldsymbol{\xi}_t \right) = 2(\boldsymbol{w} - \boldsymbol{e}_1) - \frac{2}{\sqrt{d}} \boldsymbol{V}^\top \bar{\boldsymbol{\xi}},$$

where

$$\bar{\boldsymbol{\xi}} = \frac{1}{m} \sum_{t=1}^m \boldsymbol{\xi}_t \sim N\left(\boldsymbol{0}, \frac{\sigma^2}{m} \boldsymbol{I}\right),$$

and the reduction of variance is due to averaging i.i.d. noise variables. Furthermore, we rewrite

$$\nabla L(\boldsymbol{w}) = 2(\boldsymbol{w} - \boldsymbol{e}_1 - \boldsymbol{\zeta}), \tag{11}$$

where the noise vector $\boldsymbol{\zeta} = \frac{1}{\sqrt{d}} \boldsymbol{V}^\top \bar{\boldsymbol{\xi}}$ has distribution

$$\boldsymbol{\zeta} \sim N\left(\boldsymbol{0}, \frac{\sigma^2}{md} \boldsymbol{V}^\top \boldsymbol{V}\right) = N\left(\boldsymbol{0}, \frac{\sigma^2}{md} \boldsymbol{I}\right).$$

We also bound the error of the algorithm from above:

$$e(\boldsymbol{w}_t) = \|\boldsymbol{w}_t - \boldsymbol{e}_1\|^2 = \|\boldsymbol{w}_t\|^2 - 2\boldsymbol{w}_t^\top \boldsymbol{e}_1 + \|\boldsymbol{e}_1\|^2 \leq 2 - 2\boldsymbol{w}_t^\top \boldsymbol{e}_1 = 2(1 - w_{t,1}) = 2(1 - v_{t,1}^+ + v_{t,1}^-).$$

So it suffices to upper-bound $1 - v_{t,1}^+$ and $v_{t,1}^-$.

Consider the weights of the batch $EG^\pm$ algorithm after just a *single* trial, that is $\boldsymbol{v}_2^+$ and $\boldsymbol{v}_2^-$. Using (11) and noting that $\boldsymbol{w}_1 = \boldsymbol{0}$, and $v_{1,i}^+ = v_{1,i}^- = \frac{1}{2d}$ for all $i$, we can concisely write $v_{2,1}^+$ and $v_{2,1}^-$ as:

$$v_{2,1}^+ = \frac{e^{2\eta(1+\zeta_1)}}{Z_2}, \qquad v_{2,1}^- = \frac{e^{-2\eta(1+\zeta_1)}}{Z_2}, \qquad Z_2 = \sum_{i=1}^d e^{2\eta(\delta_{1i}+\zeta_i)} + e^{-2\eta(\delta_{1i}+\zeta_i)}, \tag{12}$$

We will now lower-bound $v_{2,1}^+$, and later use a relation which directly follows from (12):

$$v_{2,1}^- = e^{-4\eta(1+\zeta_1)} v_{2,1}^+. \tag{13}$$

---

14. We can also provide an upper bound on the error of the online version of $EG^\pm$ for an arbitrary input matrix $\boldsymbol{X}$ with fixed feature range via a standard worst-case regret analysis followed by the online-to-batch conversion (see e.g. Kivinen and Warmuth (1997)). The bound so obtained would however give a slower rate of order $O(\sqrt{\log d/(dm)})$, which still has a substantially better dependence on dimension $d$ than the lower bound from the previous section.

Using the deviation bound for zero-mean Gaussian $z \sim N(0, \tau^2)$, $P(|z| \geq \gamma) \leq 2 \exp\left\{-\frac{\gamma^2}{2\tau^2}\right\}$, we get that for any $i = 1, \ldots, d$,

$$P(|\zeta_i| \geq 1/4) \leq 2 \exp\left\{-\frac{md\gamma^2}{32\sigma^2}\right\}.$$

Taking the union bound over $i = 1, \ldots, i$, we have

$$P(\exists i \ |\zeta_i| \geq 1/4) \leq 2d \exp\left\{-\frac{md\gamma^2}{32\sigma^2}\right\}.$$

Denoting the probability on the right-hand side by $\delta$, we conclude that with probability at least $1 - \delta$, all noise variables $\zeta_i$ are bounded by $1/4$. Let us call this event $E$, and we condition everything that follows on the fact that $E$ happened.

Note that for any $i \geq 2$,

$$\frac{\partial v_{2,1}^+}{\partial \zeta_i} = -\frac{v_{2,1}^+}{Z_2} 2\eta \left(e^{2\eta\zeta_i} - e^{-2\eta\zeta_i}\right),$$

which is decreasing for $\zeta_i > 0$ and increasing for $\zeta_i < 0$. So, to lower-bound $v_{2,1}^+$, conditioning on event $E$, we set $\zeta_i = 1/4$ for all $i \geq 2$ in (12) ($\zeta_i = -1/4$ would result in the same value of these weights). This gives:

$$
\begin{aligned}
v_{2,1}^+ &\geq \frac{e^{2\eta(1+\zeta_1)}}{e^{2\eta(1+\zeta_1)} + e^{-2\eta(1+\zeta_1)} + (d-1)(e^{\eta/2} + e^{-\eta/2})} \\
&= \frac{1}{1 + e^{-4\eta(1+\zeta_1)} + e^{-2\eta(1+\zeta_1)}(d-1)(e^{\eta/2} + e^{-\eta/2})} \\
&\geq \frac{1}{1 + e^{-4\eta(1-1/4)} + e^{-2\eta(1-1/4)}(d-1)(e^{\eta/2} + e^{-\eta/2})} \\
&\geq \frac{1}{1 + e^{-3\eta} + e^{-3/2\eta}2(d-1)e^{\eta/2}} \\
&\geq \frac{1}{1 + e^{-3\eta} + 2(d-1)e^{-\eta}} > \frac{1}{1 + (2d-1)e^{-\eta}}
\end{aligned}
\tag{14}
$$

This gives:

$$1 - v_{2,1}^+ \leq \frac{(2d-1)e^{-\eta}}{1 + (2d-1)e^{-\eta}} = \frac{1}{1 + e^{\eta}/(2d-1)} \leq (2d-1)e^{-\eta}.$$

To upper-bound $v_{2,1}^-$, we use (13). Conditioning on $E$,

$$v_{2,1}^- = e^{-4\eta(1+\zeta_1)}v_{2,1}^+ \leq e^{-4\eta(1+\zeta_1)} \leq e^{-2\eta}.$$

Thus, with probability at least $1 - \delta$, the error can be bounded by:

$$e(\boldsymbol{w}_2) \leq 2(1 - v_{2,1}^+ + v_{2,1}^-) \leq (2d-1)e^{-\eta} + e^{-2\eta} \leq 2de^{-\eta}$$

To get the expected error (with respect to the training data), we bound

$$
\begin{aligned}
\mathbb{E}[e(\boldsymbol{w}_2)] &= \mathbb{E}[e(\boldsymbol{w}_2)|E]P(E) + \mathbb{E}[e(\boldsymbol{w}_2)|E']P(E') \leq \mathbb{E}[e(\boldsymbol{w}_2)|E] + \delta\, \mathbb{E}[e(\boldsymbol{w}_2)|E'] \\
&\leq \mathbb{E}[e(\boldsymbol{w}_2)|E] + 2\delta = 2de^{-\eta} + 8de^{-\frac{md}{32\sigma^2}},
\end{aligned}
$$

where we used the fact that $e(\boldsymbol{w}_2) \leq 2(1 - v_{2,1}^+ + v_{2,1}^-) \leq 4$ as $v_{2,1}^+, v_{2,1}^- \in [0,1]$, and that maximizing convex function $e(\boldsymbol{w})$ give $\boldsymbol{w}$ Thus, taking sufficiently large $\eta$, we can drop the error arbitrarily close to $8de^{-\frac{md}{32\sigma^2}}$. ∎

### D.2. Upper bound for the Approximated Unnormalized Exponentiated Gradient update

We now drop the normalization of EG$^\pm$ and use the approximation[15]: $\exp(x) \approx 1 + x$. to derive the *Approximated EGU$^\pm$* algorithm (17), that is, the first-order approximation of the unnormalized Exponentiated Gradient update.

The vanilla EGU$^\pm$ algorithm keeps track of two vectors, $\boldsymbol{v}_t^+$ and $\boldsymbol{v}_t^-$, and the prediction vector is given by $\boldsymbol{w}_t = \boldsymbol{v}_t^+ - \boldsymbol{v}_t^-$. Let $\beta$ denote the initial value of weights, that is $\boldsymbol{v}_1^+ = \boldsymbol{v}_1^- = \beta\mathbf{1}$. The weights are updated according to

$$\boldsymbol{v}_{t+1}^\pm = \boldsymbol{v}_t^\pm \odot e^{\mp\eta\nabla L(\boldsymbol{w}_t)} = \beta e^{\mp\eta\sum_{j=1}^t \nabla L(\boldsymbol{w}_j)}. \tag{15}$$

At every timestamp we have $\boldsymbol{v}_t^+ \boldsymbol{v}_t^- = \beta^2$, which together with $\boldsymbol{w}_t = \boldsymbol{v}_t^+ - \boldsymbol{v}_t^-$, allows us to express $\boldsymbol{v}_t^+$ and $\boldsymbol{v}_t^-$ in terms of $\boldsymbol{w}_t$:

$$\boldsymbol{v}_t^+ = \frac{\boldsymbol{w}_t + \sqrt{\boldsymbol{w}_t^2 + 4\beta^2}}{2}, \qquad \boldsymbol{v}_t^- = \frac{-\boldsymbol{w}_t + \sqrt{\boldsymbol{w}_t^2 + 4\beta^2}}{2}, \tag{16}$$

where all operations (squaring the weights and taking square root) are done component-wise. Expanding the EGU$^\pm$ update (15) in the learning rate we get

$$\boldsymbol{v}_{t+1}^\pm = \boldsymbol{v}_t^\pm e^{\mp\eta\nabla L(\boldsymbol{w}_t)} = \boldsymbol{v}_t^\pm(1 \mp \eta\nabla L(\boldsymbol{w}_t)) + O(\eta^2).$$

Dropping the $O(\eta^2)$ term and using (16) gives

$$\boldsymbol{v}_t^+ + \boldsymbol{v}_t^- = \sqrt{\boldsymbol{w}^2 + 4\beta^2},$$

so that the update finally becomes

$$\boldsymbol{w}_{t+1} = \boldsymbol{w}_t - \eta\sqrt{\boldsymbol{w}_t^2 + 4\beta^2}\,\nabla L(\boldsymbol{w}_t), \tag{17}$$

with $\beta > 0$ being a free parameter of the algorithm. When $\beta = 0$ this update is closely related to gradient descent on the spindly network of Figure 1. We start with $\boldsymbol{w}_1 = \mathbf{0}$. Note that unlike EG$^\pm$ the update does not constrain its weights by normalizing. Nevertheless, we show that the algorithm achieves an upper bound on the error that is essentially $O(\frac{d}{\log d})$ better than the error of any rotation invariant algorithm:

**Theorem 8** *Assume $d \geq 4$ is such that $\sqrt{d}$ is an integer. Consider the Approximated EGU$^\pm$ algorithm (17) with $\beta = 1/(2d)$ and learning rate $\eta = 1/4$ and let $m \geq 8\sigma^2 \ln \frac{2d}{\delta} = \Omega(\sigma^2 \log(d/\delta))$ . With probability at least $1 - \delta$, the algorithm run for $T = 4\sqrt{d}$ steps achieves error bounded by:*

$$e(\boldsymbol{w}_{T+1}) \leq \frac{10\sigma^2 \ln \frac{2d}{\delta}}{md} + 9e^{-\frac{8}{3}\sqrt{d}+2\ln d} = O(\frac{\sigma^2 \log d}{md} + e^{-\frac{8}{3}\sqrt{d}}).$$

---

15. This approximated version of EGU was introduced in (Kivinen and Warmuth, 1997). It was also used in the normalized update PROD (Cesa-Bianchi et al., 2007).

**Proof** The proof relies on the following observations. Firstly, similarly as for $EG^\pm$, with high probability all noise variables are small. Secondly, we can interpret the weight update (17) as the gradient descent update on the weights with the *effective learning rates* $\eta\sqrt{\boldsymbol{w}_t^2 + 4\beta^2}$. Since $\beta = 1/(2d)$ and $\boldsymbol{w}_1 = \boldsymbol{0}$, these learning rates are initially small and of order $O(1/d)$. We finally show by a careful analysis of the update that the weights $w_{t,i}$ remain small for all coordinates $i$ except $i = 1$, and so do the associated learning rates. Therefore, after $T$ steps, the algorithm does not move significantly away from zero on these coordinates. Meanwhile, the weight on the first coordinate $w_{t,1}$ increases towards 1. While the initial rate of increase is small as well, it accelerates over time as $\eta\sqrt{w_{t,1}^2 + 4\beta^2}$ increases due to increasing $w_{t,1}$. Eventually $w_{T+1,1}$ gets very close to 1 after $T$ steps of the algorithm.

Using (11), $\nabla L(\boldsymbol{w}_t) = 2(\boldsymbol{w}_t - \boldsymbol{e}_1 - \boldsymbol{\zeta})$, the update (17) becomes:

$$\boldsymbol{w}_{t+1} = \boldsymbol{w}_t - 2\eta\sqrt{\boldsymbol{w}_t^2 + 4\beta^2}\,(\boldsymbol{w}_t - \boldsymbol{e}_1 - \boldsymbol{\zeta}) \tag{18}$$

We set $\beta = 1/(2d)$, $\eta = 1/4$, and the number of steps of the algorithm $T = 4\sqrt{d}$. Note that $d \geq 4$ Using the deviation bound for zero-mean Gaussian $z \sim N(0, \tau^2)$, $P(z \geq -\gamma) \leq \exp\left\{-\frac{\gamma^2}{2\tau^2}\right\}$, we get that $P(|\zeta_i| \geq \gamma) \leq 2\exp\left\{-\frac{md\gamma^2}{8\sigma^2}\right\}$. Taking the union bound we have $P(\exists i\ |\zeta_i| \geq \gamma) \leq 2d\exp\left\{-\frac{md\gamma^2}{8\sigma^2}\right\}$. Denoting the probability on the left-hand side by $\delta$, we can solve for $\gamma$:

$$\gamma = \sigma\sqrt{\frac{\ln\frac{2d}{\delta}}{md}}.$$

This means the with probability at least $1 - \delta$, $|\zeta_i| \leq \gamma$ for all $i = 1, \ldots, d$. Let us call this event $E$, and we condition everything what follows on the fact that $E$ happened. Furthermore, due to our assumption that $m$ grows at least logarithmically with $d$, $m \geq 8\sigma^2 \ln\frac{2d}{\delta}$, we have $\gamma \leq \frac{1}{\sqrt{8d}}$.

We rewrite the update (18) in terms of $\boldsymbol{s}_t = \boldsymbol{w}_t - \boldsymbol{e}_1 - \boldsymbol{\zeta}$

$$\boldsymbol{s}_{t+1} = (1 - 2\eta\sqrt{(\boldsymbol{s}_t + \boldsymbol{e}_1 + \boldsymbol{\zeta})^2 + 4\beta^2})\boldsymbol{s}_t \tag{19}$$

**The analysis for 'zero signal' direction.** Since every weights evolves independently of the other weights, we can analyze each coordinate separately. We start with any coordinate $i \geq 2$ ('zero signal' weights), for which the update becomes

$$s_{t+1,i} = (1 - 2\eta\sqrt{(s_{t,i} + \zeta_i)^2 + 4\beta^2})s_{t,i},$$

with $s_{t,1} = -\zeta_1$. Now, w.l.o.g. assume $\zeta_i < 0$ (the analysis for $\zeta_i > 0$ is analogous). This means that $s_{1,i} > 0$, and $s_{t,i}$ is positive and monotonically decreasing as long as $2\eta\sqrt{(s_{t,i} + \zeta_i)^2 + 4c^2} < 1$; this condition is ensured by noticing that

$$(s_{t,i} + \zeta_i)^2 + 4\beta^2 \leq \zeta_i^2 + d^{-2} \leq \gamma^2 + \frac{1}{4} \leq \frac{1}{2},$$

so that using learning rate $\eta < \sqrt{2}/2$ will do the trick (remind that we use $\eta = 1/4$). Since we know that $s_{t,i}$ is positive and monotonically decreasing, we can bound:

$$s_{t+1,i} \geq (1 - 2\eta\sqrt{(s_{T+1,i} + \zeta_i)^2 + 4\beta^2})s_{t,i},$$

so that

$$s_{T+1,i} \geq (1 - 2\eta\sqrt{(s_{T+1,i} + \zeta_i)^2 + 4\beta^2})^T s_{1,i} \geq (1 - 2\eta T\sqrt{(s_{T+1,i} + \zeta_i)^2 + 4\beta^2})\zeta_i,$$

where we used the Bernoulli inequality $(1 + x)^n \geq 1 + xn$. Returning to the original variable $w_{t,i} = s_{t,i} + \zeta_i$ gives

$$w_{T+1,i} \geq 2\eta T\zeta_i\sqrt{w_{T+1,i}^2 + 4\beta^2}.$$

Since we also know that $w_{T+1,i} < 0$ (because $s_{t,i} = w_{t,i} - \zeta_i$ was decreasing in $t$ with $s_{1,i} = -\zeta_i$ and $w_{1,i} = 0$), we have

$$w_{T+1,i}^2 \leq 4\eta^2 T^2 \zeta_i^2 (w_{T+1,i}^2 + 4\beta^2),$$

which can be solved for $w_{T+1,i}^2$:

$$w_{T+1,i}^2 \leq \frac{16\beta^2\eta^2 T^2 \zeta_i^2}{1 - 4\eta^2 T^2 \zeta_i^2} = \frac{1}{d^2}\frac{4\eta^2 T^2 \gamma^2}{1 - 4\eta^2 T^2 \gamma^2}$$

This expression is increasing in $\zeta_i^2$ so we can upper-bound it by

$$w_{T+1,i}^2 \leq \frac{1}{d^2}\frac{4\eta^2 T^2 \gamma^2}{1 - 4\eta^2 T^2 \gamma^2}$$

Since $\eta = 1/4$, $T = 4\sqrt{d}$ and $\gamma \leq \frac{1}{\sqrt{8d}}$, the denominator is bounded from below

$$1 - 4\eta^2 T^2 \gamma^2 \geq 1 - \frac{1}{2} \geq \frac{1}{2},$$

so that, using $\gamma = \sigma\sqrt{\frac{\ln\frac{2d}{\delta}}{md}}$,

$$w_{T+1,i}^2 \leq \frac{1}{d^2}\frac{8d \cdot \sigma^2 \ln\frac{2d}{\delta}}{md} = \frac{8\sigma^2 \ln\frac{2d}{\delta}}{md^2}.$$

Thus, the total error from 'zero-signal' coordinates is

$$\sum_{i=2}^{d} w_{T+1,i}^2 \leq \frac{8\sigma^2 \ln\frac{2d}{\delta}}{md} \tag{20}$$

We also need to show that the same amount of error comes from the first coordinate.

**Analysis for the 'signal' coordinate.** Using (19), we have for $s_{t,1} = 1 - \zeta_i - w_{t,1}$:

$$s_{t+1,1} = (1 - 2\eta\sqrt{(1 - s_{t,1} - \zeta_i)^2 + 4\beta^2})s_{t,1},$$

with $s_{1,1} = 1 - \zeta_i$. As before, we note that $s_{t,i}$ is decreasing in $t$, as long as $2\eta\sqrt{(1 - s_{t,1} - \zeta_i)^2 + 4\beta^2} < 1$. However, this condition is satisfied for our choice of of $\eta = 1/4$, because

$$2\eta\sqrt{(1 - s_{t,1} - \zeta_i)^2 + 4\beta^2} \leq 2\eta\sqrt{(1 - \zeta_i)^2 + 4\beta^2} \leq \frac{1}{2}\sqrt{1 + 2|\zeta_i| + \zeta_i^2 + d^{-2}}$$

$$\leq \frac{1}{2}\sqrt{1 + \frac{1}{\sqrt{2d}} + \frac{9}{8d^2}} \overset{d \geq 4}{\leq} \frac{1}{2}\sqrt{1 + \frac{1}{8} + \frac{9}{128}} < 1.$$

Now, we need to carefully analyze the update. Initially $s_{t,1}$ decreases slowly, as the square root term is essentially of order $1/d$. At some point, however, $s_{t,1}$ falls below a certain constant (say, $s_{t,1} = 1/2$), and the square root term is of order $O(1)$, and the convergence becomes exponential.

First, to simplify analysis we simply denote $s_{t,i}$ by $s_t$; moreover, define $r = 1 - \zeta_i$, so that the update becomes

$$s_{t+1} = (1 - 2\eta\sqrt{(r - s_t)^2 + 4\beta^2}s_t, \qquad s_1 = r, \tag{21}$$

Already after the first iteration,

$$s_2 = (1 - 1/2d^{-1})s_t = \frac{2d - 1}{2d}r,$$

so that $(r - s_2)^2 = 1/(4d^2)$ becomes comparable with $4\beta^2 = 1/d^2$ term. So we can drop the $4\beta^2$ term from the square root in (21) and upper bound

$$s_{t+1} \leq (1 - 2\eta(r - s_t))s_t \tag{22}$$

To get some insight into this expression we solve the corresponding differential equation:

$$\dot{s} = -2\eta(r - s)s,$$

which give:

$$\frac{s_t}{r - s_t} = Ce^{-\eta rt} \implies s_t = \frac{r}{1 + Ce^{2\eta rt}}.$$

Inspired by this we will bound $\frac{s_t}{r - s_t}$. From (22) we get:

$$r - s_{t+1} \geq r - (1 - 2\eta(r - s_t)s_t = (1 + 2\eta s_t)(r - s_t),$$

so that:

$$\frac{s_{t+1}}{r - s_{t+1}} \leq \underbrace{\left(\frac{1 - 2\eta(r - s_t)}{1 + 2\eta s_t}\right)}_{=:A_t} \left(\frac{s_t}{r - s_t}\right).$$

We will now bound $A_t$ independent of $s_t$. To this end, note that $A_t$ is maximized when $s_t = r$. Indeed,

$$A_t = \frac{1 + 2\eta s_t - 2\eta r}{1 + 2\eta s_t} = 1 - \frac{2\eta r}{1 + 2\eta s_t} \leq 1 - \frac{2\eta r}{1 + 2\eta r} = \frac{1}{1 + 2\eta r}.$$

This way, we get an upper bound:

$$\frac{s_{T+1}}{r - s_{T+1}} \leq (1 + 2\eta r)^{-(T-1)}\frac{s_2}{r - s_2} = (2d - 1)(1 + 2\eta r)^{-(T-1)},$$

or by solving for $s_{T+1}$,

$$s_{T+1} \leq \frac{r}{1 + (2d - 1)^{-1}(1 + 2\eta r)^{T-1}} \leq r(2d - 1)(1 + 2\eta r)^{-(T-1)}.$$

The expression above is decreasing in $r$ for $T \geq 4$ (can be verified by computing the derivative), so we we will upper-bound it by lower-bounding $r$, that is $r = 1 - \zeta_i \geq 1 - \frac{1}{\sqrt{8d}}$. Taking $\eta = 1/4$ and using $1 - \frac{1}{\sqrt{8d}} \overset{d \geq 4}{\geq} 1 - \frac{1}{4\sqrt{2}}$, we get $1 + 2\eta r \geq \frac{3}{2} - \frac{1}{8\sqrt{2}} > e^{1/3}$ (checked numerically). Therefore,

$$s_{T+1} \leq (2d - 1)e^{-(T-1)/3} \leq e^{1/3 + \ln 2}e^{-T/3 + \ln d} \leq 3e^{-T/3 + \ln d}.$$

Using the fact that $T = 4\sqrt{d}$, we get

$$s_{T+1} \leq 3e^{-\frac{4}{3}\sqrt{d} + \ln d}$$

To bound $(1 - w_{T+1,1})^2$ we use

$$(1 - w_{T+1,1})^2 = (s_{T+1,1} + \zeta_1)^2 \leq 2s_{T+1,1}^2 + 2\zeta_1^2 \leq 9e^{-\frac{8}{3}\sqrt{d} + 2\ln d} + \frac{2\sigma^2 \ln \frac{2d}{\delta}}{md}. \qquad (23)$$

**Bound the error of Approximated EGU$^\pm$ algorithm** The final error of the algorithm is obtained by summing (20) and (23):

$$\|\boldsymbol{w}_{T+1} - \boldsymbol{e}_1\|^2 \leq 9e^{-\frac{8}{3}\sqrt{d} + 2\ln d} + \frac{2\sigma^2 \ln \frac{2d}{\delta}}{md} + \frac{8\sigma^2 \ln \frac{2d}{\delta}}{md}$$
$$= \frac{10\sigma^2 \ln \frac{2d}{\delta}}{md} + 9e^{-\frac{8}{3}\sqrt{d} + 2\ln d}.$$

$\blacksquare$

### D.3. Upper bound for the spindly network

In this section, we upper-bound the error of the spindly network defined in Figure 1, showing essentially the same bound (up to constants) as for the Approximated EGU$^\pm$. We note that essentially the same algorithm has been analyzed by Vaskevicius et al. (2019), giving the same bound $O(\frac{\sigma^2 \log d}{md})$ under the Restricted Isometry Property (RIP) assumption.

**Theorem 9** *Assume $d \geq 4$ is such that $\sqrt{d}$ is an integer. Consider the spindly network given in Figure 1 trained with gradient descent, with weights initialized as $\boldsymbol{u} = \sqrt{2/d}\boldsymbol{1}$ and $\boldsymbol{v} = \boldsymbol{0}$, and the learning rate set to $\eta = 1/4$. Let $m \geq 8\sigma^2 \ln \frac{2d}{\delta} = \Omega(\sigma^2 \log(d/\delta))$ . With probability at least $1 - \delta$, the algorithm run for $T = 4\sqrt{d}$ steps achieves error bounded by:*

$$e(\boldsymbol{w}_{T+1}) \leq \frac{33\sigma^2 \ln \frac{2d}{\delta}}{4md} + 16e^{-\frac{8}{3}\sqrt{d} + 2\ln d} = O\left(\frac{\sigma^2 \log d}{md} + e^{-\frac{8}{3}\sqrt{d}}\right)$$

**Proof** The spindly network predicts with $\boldsymbol{w}_t$ given by $\boldsymbol{w}_t = \boldsymbol{u}_t \odot \boldsymbol{v}_t$, and both $\boldsymbol{u}_t$ and $\boldsymbol{v}_t$ are updated with the gradient descent algorithm:

$$\boldsymbol{u}_{t+1} = \boldsymbol{u}_t - \eta \nabla_{\boldsymbol{u}_t} L(\boldsymbol{w}_t), \qquad \boldsymbol{v}_{t+1} = \boldsymbol{v}_t - \eta \nabla_{\boldsymbol{v}_t} L(\boldsymbol{w}_t)$$

Using (11) and the chain rule, $\nabla_{\boldsymbol{u}_t} L(\boldsymbol{w}_t) = 2(\boldsymbol{w}_t - \boldsymbol{e}_1 - \boldsymbol{\zeta}) \odot \boldsymbol{v}_t$ and $\nabla_{\boldsymbol{v}_t} L(\boldsymbol{w}_t) = 2(\boldsymbol{w}_t - \boldsymbol{e}_1 - \boldsymbol{\zeta}) \odot \boldsymbol{u}_t$, so that

$$\boldsymbol{u}_{t+1} = \boldsymbol{u}_t - 2\eta(\boldsymbol{w}_t - \boldsymbol{e}_1 - \boldsymbol{\zeta}) \odot \boldsymbol{v}_t, \qquad \boldsymbol{v}_{t+1} = \boldsymbol{v}_t - 2\eta(\boldsymbol{w}_t - \boldsymbol{e}_1 - \boldsymbol{\zeta}) \odot \boldsymbol{u}_t \qquad (24)$$

Let us introduce two vectors, $\boldsymbol{v}_t^+$ and $\boldsymbol{v}_t^-$, given by:

$$\boldsymbol{v}_t^+ = \frac{1}{4}(\boldsymbol{u}_t + \boldsymbol{v}_t)^2, \qquad \boldsymbol{v}_t^- = \frac{1}{4}(\boldsymbol{u}_t - \boldsymbol{v}_t)^2,$$

(square applied coordinatewise) and note that

$$\boldsymbol{w}_t = \boldsymbol{u}_t \cdot \boldsymbol{v}_t = \boldsymbol{v}_t^+ - \boldsymbol{v}_t^-.$$

Subtracting and adding equations in (24), followed by squaring both sides gives

$$\boldsymbol{v}_{t+1}^+ = (1 - 2\eta(\boldsymbol{w}_t - \boldsymbol{e}_1 - \boldsymbol{\zeta}))^2 \odot \boldsymbol{v}_t^+$$
$$\boldsymbol{v}_{t+1}^- = (1 + 2\eta(\boldsymbol{w}_t - \boldsymbol{e}_1 - \boldsymbol{\zeta}))^2 \odot \boldsymbol{v}_t^- \tag{25}$$

We initialize the algorithm as follows:

$$\boldsymbol{u}_1 = \sqrt{\frac{2}{d}}, \qquad \boldsymbol{v}_1 = \boldsymbol{0},$$

so that $\boldsymbol{v}_1^+ = \boldsymbol{v}_1^- = \frac{1}{2d}\boldsymbol{1}$ and $\boldsymbol{w}_1 = \boldsymbol{0}$, similarly as in the Approximated EGU$^\pm$. We set $\eta = \frac{1}{8}$, $T = 4\sqrt{d}$, and use the same assumption as in the previous section, that is $m \geq 8\sigma^2 \ln \frac{2d}{\delta}$, which implies that with probability at least $1 - \delta$, $|\zeta_i| \leq \gamma$ for all $i = 1, \ldots, d$, where

$$\gamma = \sigma\sqrt{\frac{\ln \frac{2d}{\delta}}{md}} \leq \frac{1}{\sqrt{8d}}. \tag{26}$$

As before, we denote the high probability event above as $E$, and we condition everything what follows on the fact that $E$ happened. We will also assume that $d \geq 9$.

**The analysis for 'zero signal' direction.**  As before, we can analyze each coordinate separately. We start with any coordinate $i \geq 2$, for which the update (25) becomes

$$v_{t+1,i}^\pm = (1 \mp 2\eta(w_{t,i} - \zeta_i))^2 v_{t,i}^\pm \tag{27}$$

W.l.o.g. assume $\zeta_i > 0$ (the analysis for $\zeta_i < 0$ is analogous). We will first prove by induction on $t$ that $0 \leq w_{t,i} \leq \zeta_i$ for all $t = 1, \ldots, T + 1$. Since $w_{1,i} = 0$ and $\zeta_i > 0$, it clearly holds for $t = 1$. Now, assume that it holds for iterations $1, \ldots, t$ and we prove that it also holds for $t + 1$. We have:

$$\begin{aligned}
w_{t+1,i} &= v_{t+1,i}^+ - v_{t+1,i}^- \\
&= (1 - 2\eta(w_{t,i} - \zeta_i))^2 v_{t,i}^+ - (1 + 2\eta(w_{t,i} - \zeta_i))^2 v_{t,i}^- \\
&= (1 + 4\eta^2(w_{t,i} - \zeta_i)^2)w_{t,i} - 4\eta(w_{t,i} - \zeta_i)(v_{t,i}^+ + v_{t,i}^-) \geq 0, \tag{28}
\end{aligned}$$

because $w_{t,i} - \zeta_i < 0$ and $w_{t,i} \geq 0$ from the induction assumption, while $v_{t,i}^+, v_{t,i}^- \geq 0$ from their definitions.

Now, since $-\zeta_i \leq w_{q,i} - \zeta_i \leq 0$ for $q = 1, \ldots, t$ from the inductive assumption, $\eta = \frac{1}{8}$ and $|\zeta_i| \leq \gamma \leq \frac{1}{\sqrt{8d}}$ from (26), we have $0 \geq 2\eta(w_{q,i} - \zeta_i) \geq -\frac{1}{4\sqrt{8d}}$, so that

$$(1 + 2\eta(w_{q,i} - \zeta_i))^2 < 1, \qquad (1 - 2\eta(w_{q,i} - \zeta_i))^2 > 1, \qquad q = 1, \ldots, t.$$

Thus, we see from (27) that $v_{q,i}^+$ is monotonically increasing in $q$, and $v_{q,i}^-$ is monotonically decreasing in $q$. This means that $w_{q,i} = v_{q,i}^+ - v_{q,i}^-$ is monotonically increasing in $q$. Therefore,

$$(1 + 2\eta(w_{q,i} - \zeta_i))^2 \geq (1 + 2\eta(w_{1,i} - \zeta_i))^2 = (1 - 2\eta\zeta_i)^2 \qquad \text{for all } q = 1, \ldots, t,$$

and, similarly,

$$(1 - 2\eta(w_{q,i} - \zeta_i))^2 \le (1 + 2\eta(w_{1,i} - \zeta_i))^2 = (1 + 2\eta\zeta_i)^2 \qquad \text{for all } q = 1, \dots, t.$$

This let us upper-bound $v_{t+1,i}^+$ as

$$v_{t+1,i}^+ = \prod_{q=1}^t (1 - 2\eta(w_{q,i} - \zeta_i))^2 v_{1,i}^+ \le (1 + 2\eta\zeta_i)^{2t} v_{1,i}^+ = (1 + 2\eta\zeta_i)^{2t}\frac{1}{2d},$$

and similarly lower-bound $v_{t+1,i}^-$ as

$$v_{t+1,i}^- = \prod_{q=1}^t (1 + 2\eta(w_{q,i} - \zeta_i))^2 v_{1,i}^- \ge (1 - 2\eta\zeta_i)^{2t} v_{1,i}^+ = (1 - 2\eta\zeta_i)^{2t}\frac{1}{2d}.$$

We have

$$(1 + 2\eta\zeta_i)^{2t} = e^{2t \ln(1 + 2\eta\zeta_i)} \le e^{4t\eta\zeta_i},$$

where we used $\ln(1 + x) \le x$. Moreover, by Bernoulli's inequality,

$$(1 - 2\eta\zeta_i)^{2t} \ge 1 - 4t\eta\zeta_i.$$

This gives:

$$w_{t+1,i} = v_{t+1,i}^+ - v_{t+1,i}^- = \frac{1}{2d}\left((1 + 2\eta\zeta_i)^{2t} - (1 - 2\eta\zeta_i)^{2t}\right) \le \frac{1}{2d}(e^{4t\eta\zeta_i} - 1 + 4t\eta\zeta_i).$$

Now, note that

$$4t\eta\zeta_i \le \frac{T}{2}\gamma \le \frac{4\sqrt{d}}{2\sqrt{8d}} \le \frac{1}{\sqrt{2}},$$

Using the convexity of $e^x$, we have for $x \in [0, a]$ that

$$e^x = e^{\left(1 - \frac{x}{a}\right) \cdot 0 + \frac{x}{a} \cdot a} \le \left(1 - \frac{x}{a}\right) e^0 + \frac{x}{a} e^a = 1 + x\frac{e^a - 1}{a}.$$

Taking $x = 4t\eta\zeta_i$ and $a = \frac{1}{\sqrt{2}}$, we have

$$e^{4t\eta\zeta_i} \le 1 + 4t\eta\zeta_i\sqrt{2}(e^{1/\sqrt{2}} - 1) \le 1 + 6t\eta\zeta_i.$$

This allows us to bound

$$w_{t+1,i} \le \frac{1}{2d}(e^{4t\eta\zeta_i} - 1 + 4t\eta\zeta_i) \le \frac{1}{2d}10t\eta\zeta_i \le \frac{5T\eta\zeta_i}{d} = \frac{5}{2\sqrt{d}}\zeta_i \le \frac{5}{6}\zeta_i \le \zeta_i, \qquad (29)$$

where we used $d \ge 9$. This finishes the inductive proof that $0 \le w_{t,i} \le \zeta_i$ for all $t = 1, \dots, T+1$. However, applying (29) to $t = T$ gives

$$w_{T+1,i} \le \frac{5}{2\sqrt{d}}\zeta_i,$$

so that using (26)

$$w_{T+1,i}^2 \le \frac{25}{4d}\zeta_i^2 \le \frac{25}{4d}\gamma^2 = \frac{25\sigma^2 \ln\frac{2d}{\delta}}{4md^2}.$$

Thus, the total error from 'zero-signal' coordinates is

$$\sum_{i=2}^d w_{T+1,i}^2 \le \frac{25\sigma^2 \ln\frac{2d}{\delta}}{4md}. \qquad (30)$$

**Analysis for the 'signal' coordinate.** For $i = 1$, the update becomes:

$$v_{t+1,1}^{\pm} = (1 \mp 2\eta(w_{t,i} - 1 - \zeta_1))^2 v_{t,1}^{\pm}.$$

First, we will show by induction that $0 \leq w_{t,1} \leq 1 + \zeta_1$ for all $t = 1, \ldots, T + 1$. This is clearly true for $t = 1$ as $w_{t,1} = 0$ and $|\zeta_1| \leq \frac{1}{\sqrt{8d}}$. Assume now this is true for $q = 1, \ldots, t$, and we prove it is also true for $t + 1$. We have

$$
\begin{aligned}
w_{t+1,1} &= v_{t+1,1}^{+} - v_{t+1,1}^{-} \\
&= (1 - 2\eta(w_{t,1} - 1 - \zeta_1))^2 v_{t,1}^{+} - (1 + 2\eta(w_{t,1} - 1 - \zeta_1))^2 v_{t,1}^{-} \\
&= (1 + 4\eta^2(w_{t,1} - 1 - \zeta_1)^2) w_{t,1} - 4\eta(w_{t,1} - 1 - \zeta_1)(v_{t,1}^{+} + v_{t,1}^{-}) \\
&= (1 + 4\eta^2(w_{t,1} - 1 - \zeta_1)^2) w_{t,1} - 4\eta(w_{t,1} - 1 - \zeta_1)(w_{t,1} + 2v_{t,1}^{-}) \\
&= (1 - 2\eta(w_{t,1} - 1 - \zeta_1))^2 w_{t,1} + 8\eta(1 + \zeta_i - w_{t,1}) v_{t,1}^{-}. \quad (31)
\end{aligned}
$$

It follows from the induction assumption that both terms in the last line are nonnegative, thus $w_{t+1,1} \geq 0$. To show that $w_{t+1,1} \leq 1 + \zeta_1$, note that by inductive assumption $1 + 2\eta(w_{t,i} - 1 - \zeta_1)) \geq 1 - 2\eta(1 - \zeta_1) \geq 1 - 2\eta(1 + \gamma) = \frac{3}{4} - \frac{1}{4}\gamma \geq \frac{3}{4} - \frac{1}{4\sqrt{8d}} > 0$, and also $1 + 2\eta(w_{t,i} - 1 - \zeta_1)) \leq 1$. This means that $v_{q,1}^{-}$ is nonincreasing in $q$, and thus $v_{t,1}^{-} \leq v_{1,1}^{-} = \frac{1}{2d} \leq \frac{1}{18}$ (due to $d \geq 9$). Plugging this into (31), we can bound

$$
\begin{aligned}
w_{t+1,1} &\leq (1 - 2\eta(w_{t,1} - 1 - \zeta_1))^2 w_{t,1} + 8\eta(1 + \zeta_1 - w_{t,1})\frac{1}{18} \\
&= (1 - 2\eta(w_{t,1} - 1 - \zeta_1))^2 w_{t,1} + \frac{4}{9}\eta(1 + \zeta_1 - w_{t,1}). \quad (32)
\end{aligned}
$$

We now maximize the right-hand side of (32) with respect $w_{t,1} \in [0, 1 + \zeta_1]$. To this end, define function

$$f(x) = (1 + (a - x)/4)^2 x + (a - x)/18,$$

with $a = 1 + \zeta_1$. We get

$$f'(x) = -\frac{x}{2}(1 + (a - x)/4) + (1 + (a - x)/4)^2 - \frac{1}{18},$$

which is a convex quadratic function of $x$, so that $f(x)$ achieves its maximum in the left root of $f'(x)$. Solving $f'(x) = 0$ is equivalent to

$$\frac{3x^2}{16} - x\left(1 + \frac{a}{4}\right) + \frac{17}{18} + \frac{a}{2} + \frac{a^2}{16} = 0,$$

which left root is given by

$$x_\ell = \frac{1}{3}\left(2a + 8 - \sqrt{a^2 + 8a + \frac{51}{3}}\right).$$

Note that $a = 1 + \zeta_1 \geq 1 - \gamma > 1 - \frac{1}{\sqrt{8d}} \geq 1 - \frac{1}{3\sqrt{8}} \geq 0.88 := a_0$. One can verify that $x_\ell$ is increasing in $a$ (e.g. by inspecting the sign of the derivative of $x_\ell$ with respect to $a$), which means that

$$x_\ell \geq \frac{1}{3}\left(2a_0 + 8 - \sqrt{a_0^2 + 8a_0 + \frac{51}{3}}\right) \geq 1.59.$$

This value is to the right of range $[0, a]$, because $a \leq 1 + \gamma > 1 + \frac{1}{3\sqrt{8}} \leq 1.12$. This means that the maximum of $f(x)$ in the range $[0, a]$ is achieved for $x = 0$. This corresponds to $w_{t,1} = 1 + \zeta_i$ in (32), which gives:

$$w_{t+1,1} \leq 1 + \zeta_1,$$

which was to be shown by induction.

We now lower bound $w_{t+1,1}$. Using (31) and the proven fact that $1 + \zeta_i - w_{t,1} \geq 0$ for all $t$, we have

$$w_{t+1,1} \geq (1 - 2\eta(w_{t,1} - 1 - \zeta_1))^2 w_{t,1} = (1 + 2\eta(r - w_{t,1}))^2 w_{t,1}.$$

where we simplified the notation with $r = 1 + \zeta_1$. We further bound

$$(1 + 2\eta(r - w_{t,1}))^2 = (1 + 4\eta(r - w_{t,1}) + 4\eta^2(r - w_{t,1})^2) \geq 1 + 4\eta(r - w_{t,1}),$$

so that

$$w_{t+1,1} \geq (1 + 4\eta(r - w_{t,1}))w_{t,1}.$$

Now consider expression $Q_{t+1} = \frac{r}{w_{t+1,1}} - 1$. Clearly, $Q_{t+1}$ is decreasing $w_{t+1,1}$ so we have

$$
\begin{aligned}
Q_{t+1} &\leq \frac{r}{(1 + 4\eta(r - w_{t,1}))w_{t,1}} - 1 = \frac{Q_t + 1}{1 + 4\eta(r - w_{t,1})} - 1 \\
&= \frac{Q_t - 4\eta(r - w_{t,1})}{1 + 4\eta(r - w_{t,1})} = \frac{Q_t - 4\eta w_{t,1} Q_t}{1 + 4\eta(r - w_{t,1})} = Q_t \frac{1 - 4\eta w_{t,1}}{1 + 4\eta(r - w_{t,1})}.
\end{aligned}
$$

Now, note that

$$\frac{1 - 4\eta w_{t,1}}{1 + 4\eta(r - w_{t,1})} = 1 - \frac{4\eta r}{1 + 4\eta(r - w_{t,1})} \leq 1 - \frac{4\eta r}{1 + 4\eta r} = \frac{1}{1 + 4\eta r},$$

where we used $w_{t,1} \geq 0$. Thus we get

$$Q_{t+1} \leq \frac{1}{1 + 4\eta r} Q_t$$

for all $t = 1, \ldots, T$, which implies

$$Q_{T+1} \leq \frac{1}{(1 + 4\eta r)^{T-1}} Q_2$$

(we cannot start from $Q_1$ as it is undefined). To obtain $Q_2$, we note that

$$w_{2,1} = (1 + 2\eta r)^2 v_1^+ - (1 - 2\eta r)^2 v_1^- = \frac{1}{2d}\left((1 + 2\eta r)^2 - (1 - 2\eta r)^2\right) = \frac{4\eta r}{d}.$$

Therefore,

$$Q_2 = \frac{d}{4\eta} - 1 = 2d - 1 \leq 2d,$$

and so

$$Q_{T+1} = \frac{r - w_{T+1,1}}{w_{T+1,1}} \leq \frac{2d}{(1 + r/2)^{T-1}},$$

or, equivalently,

$$r - w_{T+1,1} \le r \frac{\frac{2d}{(1+r/2)^{T-1}}}{1 + \frac{2d}{(1+r/2)^{T-1}}} \le r \frac{2d}{(1+r/2)^{T-1}}.$$

We can bound $r = 1 + \zeta_1 \ge 1 - \gamma \ge 1 - \frac{1}{\sqrt{8d}} \ge 1 - \frac{1}{3\sqrt{8}} \ge 0.88$ and similarly $r \le 1 + \frac{1}{3\sqrt{8}} \le 1.12$. This gives $1 + r/2 \ge 1.44 \ge e^{1/3}$, so that

$$r - w_{T+1,1} \le 2.24 d e^{-1/3(T-1)} = 2.24 e^{1/3} e^{-4/3\sqrt{d} + \ln d} \le 4 e^{-4/3\sqrt{d} + \ln d}$$

To bound $(1 - w_{T+1,1})^2$ we use

$$(1 - w_{T+1,1})^2 = (r - w_{T+1,i} - \zeta_1)^2 \le 2(r - w_{T+1,1})^2 + 2\zeta_1^2 \le 16 e^{-8/3\sqrt{d} + 2\ln d} + \frac{2\sigma^2 \ln \frac{2d}{\delta}}{md}, \quad (33)$$

where in the last line we used (26).

**Bound the error of the Spindly network.**   The final error of the algorithm is obtained by summing (30) and (33):

$$\|\boldsymbol{w}_{T+1} - \boldsymbol{e}_1\|^2 \le 16 e^{-8/3\sqrt{d} + 2\ln d} + \frac{2\sigma^2 \ln \frac{2d}{\delta}}{md} + \frac{25\sigma^2 \ln \frac{2d}{\delta}}{4md}$$

$$= \frac{33\sigma^2 \ln \frac{2d}{\delta}}{4md} + 16 e^{-\frac{8}{3}\sqrt{d} + 2\ln d}.$$

$\blacksquare$

### D.4. Upper bound for the priming method

The priming method operates as follows (Warmuth and Amid, 2023): First, it computes the least squares estimator $\widehat{\boldsymbol{w}}_{LS} = (\boldsymbol{X}^\top \boldsymbol{X}) \boldsymbol{X}^\top \boldsymbol{y}$. Then, it scales each column of $\boldsymbol{X}$ (each feature) by the corresponding coordinate of $\widehat{\boldsymbol{w}}_{LS}$, resulting in a new matrix $\widetilde{\boldsymbol{X}} = \boldsymbol{X} \operatorname{diag}(\widehat{\boldsymbol{w}}_{LS})$. Next, it calculates the Ridge Regression solution $\widetilde{\boldsymbol{w}}_{RR}$ using the new inputs $\widetilde{\boldsymbol{X}}$ and an appropriate regularization constant $\lambda$. The final priming predictor $\widehat{\boldsymbol{w}}'$ is obtained by by multiplying it by the coordinates of $\widehat{\boldsymbol{w}}_{LS}$, that is $\widehat{\boldsymbol{w}}' = \operatorname{diag}(\widehat{\boldsymbol{w}}_{LS}) \widetilde{\boldsymbol{X}}$. In the proof we rewrite the priming predictor $\widehat{\boldsymbol{w}}'$ as a regularized least-squares solution, with the square regularizer based on a matrix $\lambda \operatorname{diag}(\widehat{\boldsymbol{w}}_{LS})^{-2}$. Note that the regularization strength is amplified along directions where the coordinates of $\widehat{\boldsymbol{w}}_{LS}$ are small in magnitude, effectively biasing the algorithm towards sparse solutions. The proof carefully bounding the expected error of such a predictor.

**Theorem 10** *Consider the priming method equipped with $\lambda = \sigma^2 \sqrt{d}$. The expected error can be bounded by:*

$$e(\widehat{\boldsymbol{w}}') \le \frac{17\sigma^2}{md} + \frac{32\sigma^4}{m^2 d} + \frac{4\sigma e^{-md/(8\sigma^2)}}{\sqrt{2\pi md}}.$$

Note that this upper bound is by a factor of $O(\log d)$ *better* (assuming $\sigma^2 = O(1)$) than the upper bound for Approximated EGU$^\pm$ (and thus by a factor of $O(d)$ better than the error of any rotation invariant algorithm). However we don't know whether such an improved upper bound is also possible for Approximated EGU$^\pm$.

**Proof** We start with rewriting the priming method into a form which is easier to analyze. Let $\widehat{w}_{LS} = (X^\top X)X^\top y$ be the least-squares solution, which induces a diagonal weight matrix $W = \text{diag}(\widehat{w}_{LS}$. The new (rescaled) input matrix is then given by $\widetilde{X} = XW$. The ridge regression solution on the new inputs with regularization constant $\lambda$ is then $\widetilde{w} = (\widetilde{X}^\top \widetilde{X} + \lambda I_d)^{-1}\widetilde{X}^\top y$. Finally the output of the algorithm is $\widehat{w}' = W\widetilde{w}$. We thus have

$$
\begin{aligned}
\widehat{w}' &= W(\widetilde{X}^\top \widetilde{X} + \lambda I_d)^{-1}\widetilde{X}^\top y \\
&= W(WX^\top XW + \lambda I_d)^{-1}WXy \\
&= \left(W^{-1}(WX^\top XW + \lambda I_d)W^{-1}\right)^{-1} Xy \\
&= \left(X^\top X + \lambda W^{-2}\right)^{-1} Xy
\end{aligned}
$$

(in any of the elements of $\widehat{w}_{LS}$ is zero, take the limit of the expression above). Thus, $\widehat{w}'$ is a regularized least-square solution with the quadratic regularization matrix $\lambda W^{-2}$:

$$
\widehat{w}' = \underset{\widehat{w}}{\text{argmin}}\left\{\frac{1}{2}\|y - X\widehat{w}\|^2 + \frac{\lambda}{2}\widehat{w}^\top W^{-2}\widehat{w}\right\}.
$$

Note that the regularization strength is amplified along directions where the coordinates of $\widehat{w}_{LS}$ are small in magnitude, effectively biasing the algorithm towards sparse solutions.

We now specialize the priming predictor to our setup. We have

$$
X^\top X = nI_d, \quad \text{and} \quad X^\top y = X^\top Xe_1 + X^\top \xi = ne_1 + \sqrt{n}\zeta,
$$

where $\zeta = n^{-1/2}X^\top \xi$. Being a linear function of $\xi$, $\zeta$ has a normal distribution with parameters that can be obtained from:

$$
\mathbb{E}[\zeta] = n^{-1/2}X^\top \mathbb{E}[\xi] = 0, \qquad \mathbb{E}[\zeta\zeta^\top] = n^{-1}X^\top \mathbb{E}[\xi\xi]X = n^{-1}\sigma^2 X^\top X = \sigma^2 I_d,
$$

that is $\zeta \sim \mathcal{N}(0, \sigma^2 I_d)$. This gives us the expression for $\widehat{w}_{LS}$:

$$
\widehat{w}_{LS} = (X^\top X)X^\top y = n^{-1}(ne_1 + \sqrt{n}\zeta) = e_1 + n^{-1/2}\zeta,
$$

as well as for $\widehat{w}'$:

$$
\widehat{w}' = \left(X^\top X + \lambda W^{-2}\right)^{-1} Xy = (nI_d + \lambda W^{-2})^{-1}(ne_1 + \sqrt{n}\zeta)
$$

We will analyze $\widehat{w}'$ separately for each coordinate $i = 1, \ldots, d$. To simplify the notation, let $w_i$ be the $i$-th coordinate of $\widehat{w}'$, and let $v_i$ be the $i$-th coordinate of $\widehat{w}_{LS}$. The error of $\widehat{w}'$ is given by:

$$
e(\widehat{w}') = \|\widehat{w}' - e_1\|^2 = (w_1 - 1)^2 + \sum_{i=2}^{d} w_i^2.
$$

As $\boldsymbol{W} = \mathrm{diag}(\widehat{\boldsymbol{w}}_{LS})$ is diagonal, we have for any $i > 1$

$$w_i = \frac{\sqrt{n}\zeta_i}{n + \lambda v_i^{-2}} = \frac{\sqrt{n}v_i^2\zeta_i}{nv_i^2 + \lambda} = \frac{1}{\sqrt{n}}\frac{\zeta_i^3}{\zeta_i^2 + \lambda},$$

where we used $v_i = n^{-1/2}\zeta_i$ for $i > 1$. Thus the error on the $i$-th coordinate is

$$w_i^2 = \frac{1}{n}\frac{\zeta_i^6}{(\zeta_i^2 + \lambda)^2} \le \frac{1}{n}\frac{\zeta_i^6}{\lambda^2}.$$

Taking expectation over $\zeta_i \sim \mathcal{N}(0, \sigma^2)$ and using $\mathbb{E}[\zeta_i^6] = 15\sigma^6$,

$$\mathbb{E}[w_i^2] \le \frac{15\sigma^6}{n\lambda^2}.$$

We now switch to coordinate $i = 1$. We have $v_1 = 1 + n^{-1/2}\zeta_1 = n^{-1}(n + \sqrt{n}\zeta_1)$ so that

$$w_1 = \frac{n + \sqrt{n}\zeta_1}{n + \lambda v_1^2} = \frac{v_1^2(n + \sqrt{n}\zeta_1)}{v_1^2 n + \lambda} = \frac{n^{-2}(n + \sqrt{n}\zeta_1)^3}{n^{-1}(n + \sqrt{n}\zeta_1)^2 + \lambda} = \frac{(n + \sqrt{n}\zeta_1)^3}{n(n + \sqrt{n}\zeta_1)^2 + n^2\lambda}.$$

The error on the first coordinate is thus

$$(w_1 - 1)^2 = \left(\frac{(n + \sqrt{n}\zeta_1)^3 - n(n + \sqrt{n}\zeta_1)^2 - n^2\lambda}{n(n + \sqrt{n}\zeta_1)^2 + n^2\lambda}\right)^2$$

$$= \left(\frac{\sqrt{n}\zeta_1(n + \sqrt{n}\zeta_1)^2 - n^2\lambda}{n(n + \sqrt{n}\zeta_1)^2 + n^2\lambda}\right)^2.$$

Using $(a + b) \le 2a^2 + 2b^2$ in the numerator and $(a + b) \ge a^2 + b^2$ for $a, b \ge 0$ in the denominator, we bound

$$(w_1 - 1)^2 \le 2\frac{n\zeta_1^2(n + \sqrt{n}\zeta_1)^4 + n^4\lambda^2}{(n(n + \sqrt{n}\zeta_1)^2 + n^2\lambda)^2} \le \frac{2n\zeta_1^2(n + \sqrt{n}\zeta_1)^4}{n^2(n + \sqrt{n}\zeta_1)^4} + \frac{2n^4\lambda^2}{n^2(n + \sqrt{n}\zeta_1)^4 + n^4\lambda^2}$$

$$= \frac{2\zeta_1^2}{n} + \frac{2\lambda^2}{(\sqrt{n} + \zeta_1)^4 + \lambda^2}.$$

We now take expectation with respect $\zeta_1$ and get:

$$\mathbb{E}[(w_1 - 1)^2] = \frac{2\sigma^2}{n} + \mathbb{E}\left[2\lambda^2((\sqrt{n} + \zeta_1)^4 + \lambda^2)^{-1}\right].$$

The second term requires more work. Using $f(\zeta_1) = 2\lambda^2((\sqrt{n} + \zeta_1)^4 + \lambda^2)^{-1}$:

$$\mathbb{E}[f(\zeta_1)] = \mathbb{E}[f(\zeta_1)|\zeta_1 \le c]\,P(\zeta_1 \le c) + \mathbb{E}[f(\zeta_1)|\zeta_1 \ge c]\,P(\zeta_1 \ge c))$$

$$\le P(\zeta_1 \le c)\max_{\zeta_1 \le c}\{f(\zeta_1)\} + \max_{\zeta_1 \ge c}\{f(\zeta_1)\}.$$

We take $c = -\sqrt{n}/4$ and use a bound $P(Z < -t) \le \frac{\exp\{-t^2/2\}}{\sqrt{2\pi}t}$ to get

$$P(\zeta_1 \le -\sqrt{n}/4) = P(Z \le -\sqrt{n}/(4\sigma)) \le \frac{2\sigma e^{-n/(32\sigma^2)}}{\sqrt{2\pi n}}.$$

Moreover,

$$\max_{\zeta_1 \leq c}\{f(\zeta_1)\} \overset{\zeta_1 = -\sqrt{n}}{=} 2, \qquad \max_{\zeta_1 \geq c}\{f(\zeta_1)\} \overset{\zeta_1 = c}{=} \frac{2\lambda^2}{\left(\frac{3}{4}\right)^4 n^2 + \lambda^2} \leq \frac{2\lambda^2}{\left(\frac{3}{4}\right)^4 n^2} \leq \frac{7\lambda^2}{n^4}$$

Thus,

$$\mathbb{E}[(w_1 - 1)^2] \leq \frac{2\sigma^2}{n} + \frac{7\lambda^2}{n^2} + \frac{4\sigma e^{-n/(32\sigma^2)}}{\sqrt{2\pi n}}.$$

Taking it all together, we have

$$\mathbb{E}[\|\widehat{\boldsymbol{w}}' - \boldsymbol{e}_1\|^2] \leq (d-1)\frac{15\sigma^6}{n\lambda^2} + \frac{2\sigma^2}{n} + \frac{7\lambda^2}{n^2} + \frac{4\sigma e^{-n/(32\sigma^2)}}{\sqrt{2\pi n}}.$$

Without optimizing too much, we simply take $\lambda^2 = \sqrt{dn}\sigma^3 = d\sqrt{m}\sigma^3$ and get

$$\mathbb{E}[e(\widehat{\boldsymbol{w}}')] = \mathbb{E}[\|\widehat{\boldsymbol{w}}' - \boldsymbol{e}_1\|^2] \leq \frac{2\sigma^2}{n} + \frac{22\sigma^3\sqrt{d}}{n^{3/2}} + \frac{4\sigma e^{-n/(32\sigma^2)}}{\sqrt{2\pi n}}$$

$$= \frac{2\sigma^2}{md} + \frac{22\sigma^4}{m^{3/2}d} + \frac{4\sigma e^{-md/(32\sigma^2)}}{\sqrt{2\pi md}}.$$

∎

## D.5. Theorem 3 and EGU equivalences

**Proof** (Theorem 3)

For reparameterization we have

$$\dot{\boldsymbol{w}} = \frac{\partial \boldsymbol{w}}{\partial \widehat{\boldsymbol{w}}}\dot{\widehat{\boldsymbol{w}}}$$

$$= -\frac{\partial \boldsymbol{w}}{\partial \widehat{\boldsymbol{w}}}\nabla_{\widehat{\boldsymbol{w}}}L$$

$$= -\frac{\partial \boldsymbol{w}}{\partial \widehat{\boldsymbol{w}}}\left(\frac{\partial \boldsymbol{w}}{\partial \widehat{\boldsymbol{w}}}\right)^{\top}\nabla_{\boldsymbol{w}}L$$

so the preconditioner is $\frac{\partial \boldsymbol{w}}{\partial \widehat{\boldsymbol{w}}}\left(\frac{\partial \boldsymbol{w}}{\partial \widehat{\boldsymbol{w}}}\right)^{\top}$.

For MD we have

$$\dot{\boldsymbol{w}} = \frac{\partial \boldsymbol{w}}{\partial \widetilde{\boldsymbol{w}}}\dot{\widetilde{\boldsymbol{w}}}$$

$$= -\frac{\partial \boldsymbol{w}}{\partial \widetilde{\boldsymbol{w}}}\nabla_{\boldsymbol{w}}L$$

so the preconditioner is $\frac{\partial \boldsymbol{w}}{\partial \widetilde{\boldsymbol{w}}}$.

For Riemannian GD the update

$$\dot{\boldsymbol{w}} = -\boldsymbol{\Gamma}_{\boldsymbol{w}}\nabla_{\boldsymbol{w}}L$$

immediately implies the preconditioner is $\boldsymbol{\Gamma}_{\boldsymbol{w}}^{-1}$.

∎

We now analyze the implications of the theorem for EGU. EGU is defined by the mirror map (applied componentwise):

$$f(\boldsymbol{w}) = \log \boldsymbol{w}$$

This implies

$$\frac{\partial \boldsymbol{w}}{\partial \widetilde{\boldsymbol{w}}} = \mathrm{diag}(\boldsymbol{w}) \tag{34}$$

Now consider the reparameterization (applied componentwise):

$$\widehat{\boldsymbol{w}} = 2\sqrt{\boldsymbol{w}}$$

This implies

$$\frac{\partial \boldsymbol{w}}{\partial \widehat{\boldsymbol{w}}} = \tfrac{1}{2}\mathrm{diag}(\widehat{\boldsymbol{w}})$$

and therefore

$$\frac{\partial \boldsymbol{w}}{\partial \widehat{\boldsymbol{w}}} \left(\frac{\partial \boldsymbol{w}}{\partial \widehat{\boldsymbol{w}}}\right)^{\top} = \frac{\partial \boldsymbol{w}}{\partial \widetilde{\boldsymbol{w}}}$$

as in Theorem 3. This means continuous-time EGU is equivalent to gradient flow on the spindly network when the two weights are set to be equal, i.e. $w_i = u_i^2$ instead of $w_i = u_i v_i$ (note that if $\boldsymbol{u}$ and $\boldsymbol{v}$ are initialized equally then they will remain equal) (Warmuth et al., 2021).

Using (34), Theorem 3 also implies EGU is equivalent to Riemannian GD with metric

$$\boldsymbol{\Gamma}_{\boldsymbol{w}} = \mathrm{diag}(\boldsymbol{w})^{-1} \tag{35}$$

We visualize this metric in Figure 4 for a 2d parameter space. This geometry that is implicit in EGU helps to explain how the algorithm encourages weight trajectories to stay near sparse solutions.

## Appendix E. Trajectory derivations

We analyze several algorithms on the regression problem of Sections 2.3 and D. The mean loss over the training set is

$$L(\boldsymbol{w}) = \frac{1}{md}\|\boldsymbol{X}\boldsymbol{w} - \boldsymbol{y}\|^2$$

Using $\boldsymbol{X}^{\top}\boldsymbol{X} = md\boldsymbol{I}$, the gradient is

$$\nabla_{\boldsymbol{w}} L = 2(\boldsymbol{w} - \boldsymbol{w}^{\mathrm{LS}})$$

$$\boldsymbol{w}^{\mathrm{LS}} = \frac{1}{md}\boldsymbol{X}^{\top}\boldsymbol{y}$$

where $\boldsymbol{w}^{\mathrm{LS}}$ is the linear least-squares solution to which all considered algorithms converge.

The continuous EGU update can be written as

$$\dot{w}_i = -w_i \nabla_{w_i} L = 2w_i(w_i^{\mathrm{LS}} - w_i)$$

The trajectory in (4) can be directly verified to satisfy this PDE, with the initial condition $w_i(0)$ satisfied by setting

$$c_i = \tanh^{-1}\left(\frac{2w_i(0)}{w_i^{\mathrm{LS}}} - 1\right)$$

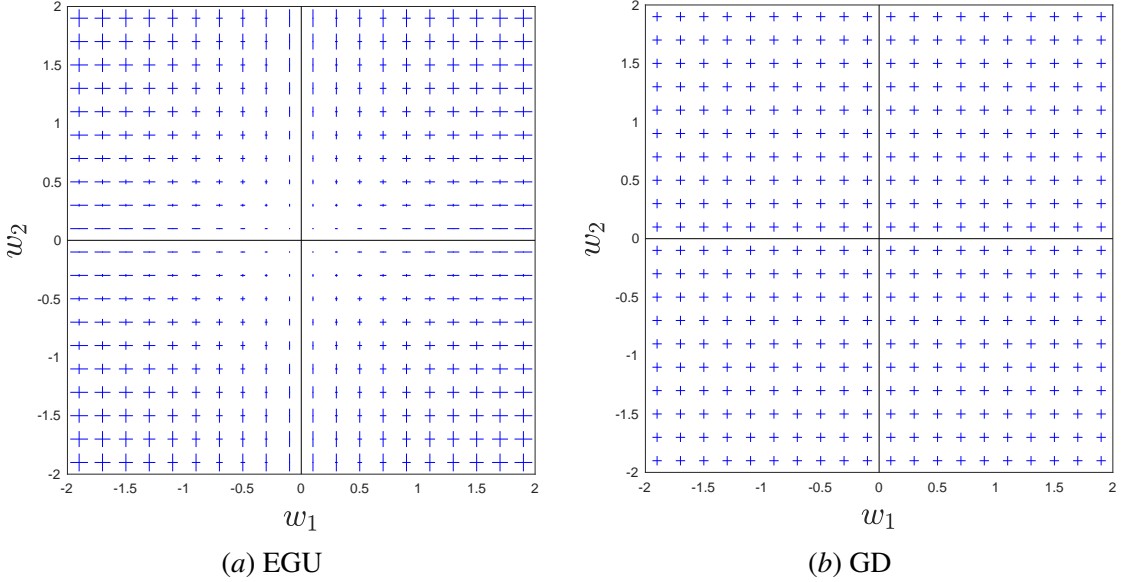

$(a)$ EGU     $(b)$ GD

**Figure 4:** (a): Metric in (35) for the Riemannian GD interpretation of EGU. (b): For comparison, the Euclidean metric implicit in GD, which is uniform and in particular rotationally symmetric. Blue lines indicate intervals of constant distance according to the respective metric.

The continuous EGU$^{\pm}$ update can be written as (Amid and Warmuth, 2020)

$$\dot{w}_i = -\sqrt{w_i^2 + 1}\,\nabla_{w_i}L = 2\sqrt{w_i^2 + 1}\,(w_i^{\mathrm{LS}} - w_i)$$

The trajectory in (5) can be directly verified to satisfy this PDE, with the initial condition $w_i(0)$ satisfied by setting

$$\tau_i = \sinh^{-1}\left(\frac{1 + w_i^{\mathrm{LS}}w_i(0)}{w_i(0) - w_i^{\mathrm{LS}}}\right) + 2t\xi\sqrt{(w_i^{\mathrm{LS}})^2 + 1}$$
$$\xi = \mathrm{sign}\left(w_i(0) - w_i^{\mathrm{LS}}\right)$$

Primed gradient flow amounts to GD under the reparameterization $\widehat{\boldsymbol{w}} = \mathrm{diag}(\boldsymbol{w}^{\mathrm{LS}})^{-1}\boldsymbol{w}$ and so by Theorem 3 the update can be written as

$$\dot{w}_i = -\left(w_i^{\mathrm{LS}}\right)^2\nabla_{w_i}L = 2\left(w_i^{\mathrm{LS}}\right)^2(w_i^{\mathrm{LS}} - w_i)$$

The trajectory in (6) can be directly verified to satisfy this PDE and the initial condition $w_i(0)$.

**A**dagrad: Continuous-time Adagrad can be written as

$$\dot{w}_i = -G_i^{-1/2}\nabla_{w_i}L = 2G_i^{-1/2}(w_i^{\mathrm{LS}} - w_i)$$
$$\dot{G}_i = \beta(\nabla_{w_i}L)^2 = 4\beta(w_i^{\mathrm{LS}} - w_i)^2$$

with preconditioner learning rate $\beta$ and $G_i(0) = \varepsilon$ a stability parameter.

To solve these coupled PDEs we begin by defining

$$\delta_i = w_i - w_i^{\text{LS}}$$

which leads to

$$\dot{\delta}_i = -2G_i^{-1/2}\delta_i$$
$$\dot{G}_i = 4\beta\delta_i^2 \tag{36}$$

Combining these two equations yields

$$\ddot{G}_i = 8\beta\delta_i\dot{\delta}_i$$
$$= -4G_i^{-1/2}4\beta\delta_i^2$$
$$= -4G_i^{-1/2}\dot{G}_i$$

which has solution

$$G_i = \frac{16}{k_i^2}\left(W\left(-e^{-k_i(t+\ell_i)}\right) + 1\right)^2$$
$$\dot{G}_i = -\frac{32}{k_i}W\left(-e^{-k_i(t+\ell_i)}\right) \tag{37}$$

Using $G_i(0) = \varepsilon$ and $\dot{G}_i(0) = 4\beta\left(w_i^{\text{LS}} - w_i(0)\right)^2$ allows to solve for the constants $k_i$ and $\ell_i$:

$$k_i = \frac{8}{\beta(w_i^{\text{LS}} - w_i(0))^2 + 2\sqrt{\varepsilon}}$$
$$\ell_i = \frac{1}{k_i} - \frac{1}{k_i}\log\left(1 - \frac{k_i\sqrt{\varepsilon}}{4}\right) - \frac{\sqrt{\varepsilon}}{4}$$

Substituting (36) and (37) gives the corresponding expression for $w_i$, matching (7):

$$w_i = w_i^{\text{LS}} + \delta_i$$
$$= w_i^{\text{LS}} + \text{sign}\left(w_i(0) - w_i^{\text{LS}}\right)\frac{\sqrt{\dot{G}_i}}{2\sqrt{\beta}}$$
$$= w_i^{\text{LS}} - \text{sign}\left(w_i^{\text{LS}} - w_i(0)\right)\sqrt{-\frac{8}{\beta k_i}W\left(-e^{-k_i(t+\ell_i)}\right)}$$

**I**ncremental priming and Burg MD: We also consider an incremental version of priming where the learned weights are continuously transferred into the priming vector rather than only once at the end of a pre-training phase. Specifically, we begin with the predictive model

$$\widehat{y}_t(\boldsymbol{x}) = (\boldsymbol{x} \odot \boldsymbol{p}_t)^\top \boldsymbol{w}_t$$

where $\boldsymbol{p}_t$ is the priming vector, $\boldsymbol{w}_t$ is the weight vector, and $t$ indexes iterations of the learning algorithm. The idea is to maintain $\boldsymbol{w}_t = \boldsymbol{1}$, transferring each update of $\boldsymbol{w}$ immediately into $\boldsymbol{p}$. Specifically, at each time step we make a provisional GD update

$$\widetilde{\boldsymbol{w}}_{t+1} = \boldsymbol{w}_t - \eta\nabla_{\boldsymbol{w}_t}L(\widehat{\boldsymbol{y}}_t(\boldsymbol{X}), \boldsymbol{y})$$
$$= \boldsymbol{1} + \eta\text{diag}(\boldsymbol{p}_t)\boldsymbol{X}^\top(\boldsymbol{y} - \boldsymbol{X}\boldsymbol{p}_t)$$

where the second line uses the inductive assumption $\boldsymbol{w}_t = \mathbf{1}$. This assumption holds because we immediately transfer the provisional update into $\boldsymbol{p}$:

$$\boldsymbol{p}_{t+1} = \boldsymbol{p}_t \odot \widetilde{\boldsymbol{w}}_{t+1}$$
$$\boldsymbol{w}_{t+1} = \mathbf{1}$$

This transfer leaves predictions at step $t+1$ unchanged because $(\boldsymbol{x} \odot \boldsymbol{p}_t)^\top \widetilde{\boldsymbol{w}}_{t+1} = (\boldsymbol{x} \odot \boldsymbol{p}_{t+1})^\top \boldsymbol{w}_{t+1}$ for all $\boldsymbol{x}$, but it leads the learning on step $t+1$ to contribute immediately to priming on future steps.

We now simplify the model by dropping the inconsequential $\boldsymbol{w}$ and directly computing the update for $\boldsymbol{p}$:

$$\widehat{y}_t(\boldsymbol{x}) = \boldsymbol{x}^\top \boldsymbol{p}_t$$
$$\boldsymbol{p}_{t+1} = \boldsymbol{p}_t \odot (1 + \eta \mathrm{diag}(\boldsymbol{p}_t) \boldsymbol{X}^\top (\boldsymbol{y} - \boldsymbol{X}\boldsymbol{p}_t))$$
$$= \boldsymbol{p}_t - \eta \mathrm{diag}(\boldsymbol{p}_t)^2 \nabla_{\boldsymbol{p}_t} L(\widehat{\boldsymbol{y}}_t(\boldsymbol{X}), \boldsymbol{y})$$

Treating $\boldsymbol{p}$ as the weight vector, we have a preconditioned GD algorithm which, by Theorem 3, is equivalent in the continuous-time case to MD with mirror map $f(\boldsymbol{p}) = -\boldsymbol{p}^{-1}$ (componentwise). This corresponds to Burg MD (Jain et al., 2007; Amid and Warmuth, 2020). (In the discrete time case, incremental priming corresponds to the "dual update" of Burg MD (Warmuth and Jagota, 1998), $\boldsymbol{p}_{t+1} = \boldsymbol{p}_t - \eta \nabla_{f(\boldsymbol{p}_t)} L$ instead of $f(\boldsymbol{p}_{t+1}) = f(\boldsymbol{p}_t) - \eta \nabla_{\boldsymbol{p}_t} L$.)

Changing notation from $\boldsymbol{p}$ back to $\boldsymbol{w}$, we can write the update as

$$\dot{w}_i = -w_i^2 \nabla_{w_i} L = 2w_i^2 (w_i^{\mathrm{LS}} - w_i)$$

This has the solution

$$w_i(t) = \frac{w_i^{\mathrm{LS}}}{W\left(\exp\left(-2\left(w_i^{\mathrm{LS}}\right)^2 (t - b_i)\right)\right) + 1}$$
$$b_i = \frac{1}{2\left(w_i^{\mathrm{LS}}\right)^2} \log W^{-1}\left(\frac{w_i^{\mathrm{LS}}}{w_i(0)} - 1\right)$$

where $W$ is Lambert's W function as above. The solution can be verified by substituting it back into the PDE.

## Appendix F. Anisotropic covariance

Theorem 2 and the analytic trajectory solutions in Section 3.2 assume spherical input covariance, meaning $\boldsymbol{X}^\top \boldsymbol{X}$ is proportional to the identity. When it is not, rotationally invariant algorithms can produce trajectories that curve toward sparse solutions under certain circumstances (Figure 5(a)). Nevertheless, Theorem 1 implies a rotationally invariant algorithm cannot perform better on sparse over non-sparse problems with a rotationally symmetric input distribution, i.e. when averaged over all rotations of $\boldsymbol{X}^\top \boldsymbol{X}$.

For a rotationally invariant algorithm on a linear problem, rotating the problem rotates the entire weight trajectory: $X \to XU^\top$ implies $w(t) \to Uw(t)$. This allows us to understand the algorithm's behavior with a fixed sparse target $v$ and rotated input $XU^\top$ by examining its behavior with a

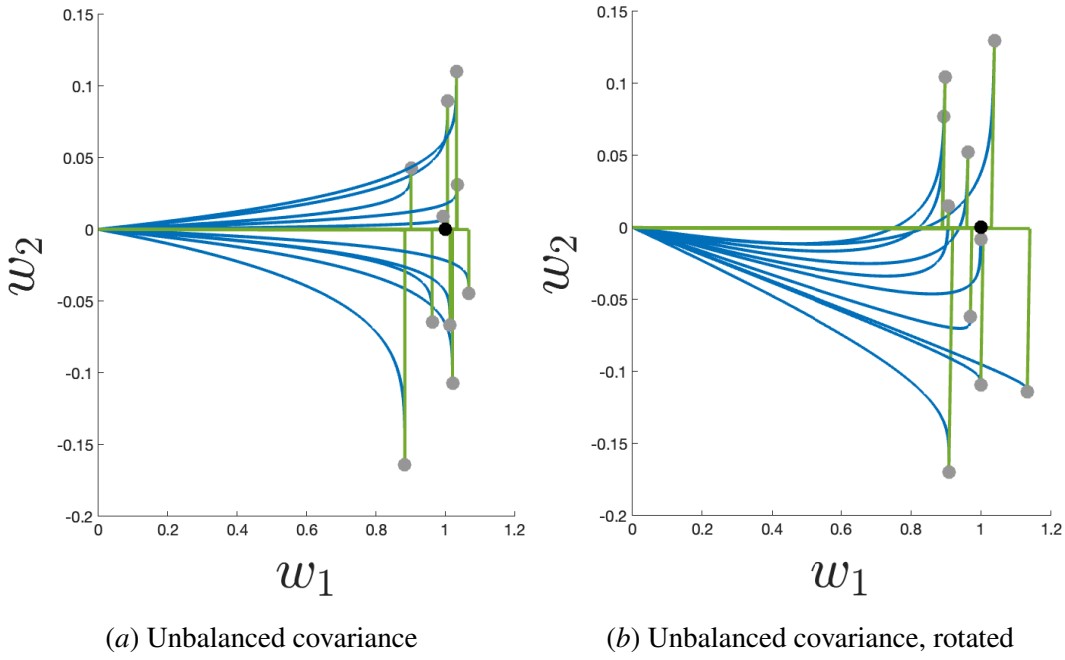

(a) Unbalanced covariance             (b) Unbalanced covariance, rotated

**Figure 5:** GD produces curved trajectories (blue) when the covariance $\boldsymbol{X}^\top\boldsymbol{X}$ is nonspherical. This can speed learning then the first principal component is aligned with the target as in (a). However, GD and other rotationally invariant algorithms cannot produce a systematic bias toward sparsity. When the input distribution is rotated as in (b), the trajectories rotate as well such that they no longer learn the sparse target efficiently. In contrast, algorithms that are not invariant by rotations, such as EGU$^\pm$ (green), can learn sparse targets efficiently under any rotation of the input.

rotated target $Uv$ and fixed input $X$. We illustrate this in Figure 5 for a 2-dimensional problem with $X = H \operatorname{diag}(2, 1)$ so that $X^\top X$ has condition number 4. On the left, the first principal component of $\boldsymbol{X}^\top\boldsymbol{X}$ is aligned with the sparse target. GD and EGU$^\pm$ both produce trajectories that bend toward the target, but for different reasons: GD's sparsity bias depends on $\boldsymbol{X}$ while EGU$^\pm$'s does not. This is seen in the right figure where the input is rotated. Rotation invariance of GD implies its trajectories also rotate, so that it no longer learns efficiently. Thus GD has no sparsity advantage when averaging over all rotated inputs. In contrast, EGU$^\pm$ shows a sparsity bias in all cases. Because it is not rotationally invariant, rotating the problem does not rotate its trajectories (they are altered but to a much lesser degree).

## Appendix G. Fashion MNIST experiment details

As with any real dataset, the input distribution of fMNIST is not rotationally symmetric. Therefore one could trivially cheat this task with a rotationally invariant algorithm, by hard-coding a lookup table of the test set. If $\boldsymbol{X}^\top\boldsymbol{X}$ is the (known, nonspherical) covariance of the original training set, then when presented with rotated training data $\boldsymbol{Z} = \boldsymbol{X}\boldsymbol{U}^\top$, the algorithm can infer the unknown $\boldsymbol{U}$ to high precision by solving $\boldsymbol{Z}^\top\boldsymbol{Z} = \boldsymbol{U}\boldsymbol{X}^\top\boldsymbol{X}\boldsymbol{U}^\top$. It then un-rotates $\boldsymbol{z}_{\text{te}}$ and uses the lookup table. Such a bespoke algorithm is of course not of interest; our question is whether standard rotation invariant algorithms based on gradient descent can do anything similar, or whether they are limited by their rotation invariance just as they are in the idealized symmetric-input setting.

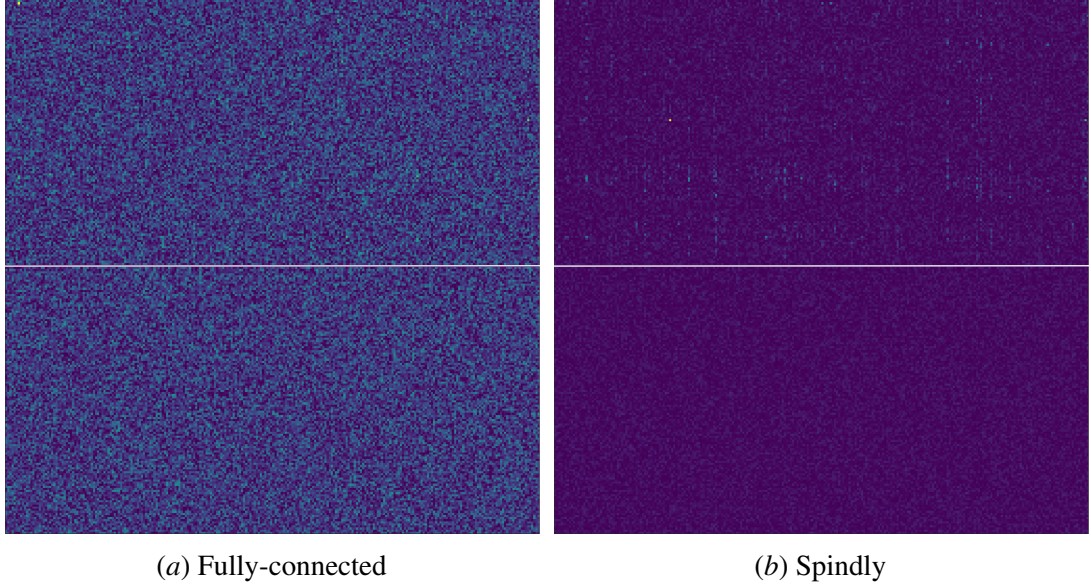

(*a*) Fully-connected          (*b*) Spindly

**Figure 6:** The weights of the first layer when trained with images augmented with noise. The top slice corresponds to the image feature weights and the bottom slice corresponds to the noise feature weights.

In our experiments, we use a constant learning rate (which we tune for each case). We use the full batch of 60000 training examples and train each network for 5000 epochs. We first provide some visualization of the weights for the noisy case where each example is augmented with unifrom noise. Figure 6 shows a subset of the weights for each network where the top slice corresponds to the image feature weights and the bottom slice corresponds to the noise feature weights. For the spindly network, the average maximum absolute value of the effective weights (i.e., the product of the two weights within each spindle) for each input neuron is 0.0182 for the image weights and 0.0025 for the noise features. The difference is less drastic for the fully-connected network, where the values are 0.0627 and 0.0568, respectively.

Next, we show the results when adding extra one-hot embeddings of the labels as features. Figure 7 shows a subset of the image and label weights for each network, along with the weights corresponding to the labels at the bottom. The spindly network assigns relatively larger weights to the label features. The average maximum absolute value of the weights for each neuron is 0.7834 for the label weights, whereas the image and noise weights have values 0.0057 and 0.0025, respectively. Again, the difference between the label weights and the rest is less prominent for the fully-connected network: The average maximum absolute values are 0.4213 for label weights and 0.0604 and 0.0584 for the image and noise weights, respectively.

In summary, the network with the fully connected input layer is more easily confused by noisy features. When noisy and informative features are added then the network with a spindlified input layer finds a model that almost entirely relies on the informative features whereas the net with the fully connected input layer still makes use of original and noisy features.

**M**ore network details: Three layer fully connected net with RELU transfer functions, 10 outputs, square loss, trained with batch gradient descent for 5000 epochs, batch size 60000.

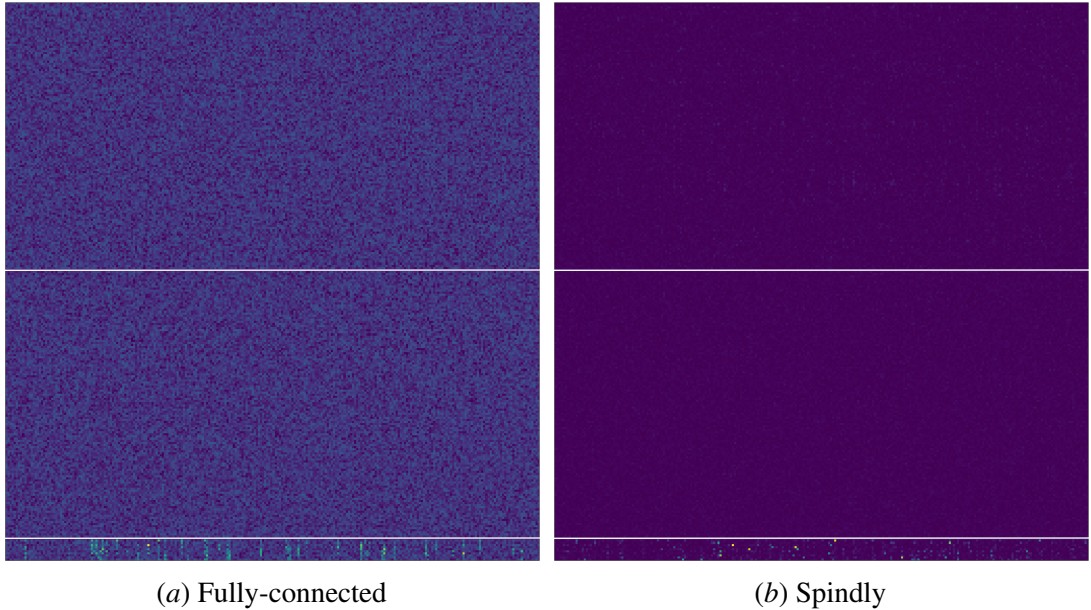

(*a*) Fully-connected         (*b*) Spindly

**Figure 7:** The weights of the first layer when trained with images augmented with noise and one-hot representation of the labels. The top slice corresponds to the image feature weights, the middle slice corresponds to the noise feature weights, and the bottom corresponds to the label weights.

Number of weights in the net with the fully connected input layer: input size 784, 256 nodes on the first two hidden layers, 10 outputs, plus biases, for a total of 269,322 weights. Adding a copy of noisy input features almost doubles the number of weights, adding $784 * 256 = 200{,}704$. The number of label weights is $10 * 256 = 2560$. It is important to note that the number of examples is much larger than the input dimension but much smaller than the total number of weights.

