# OpenReview forum: "How rotation invariant algorithms are fooled by noise on sparse targets"
_algorithmiclearningtheory.org/ALT/2025/Conference — ALT 2025_

### Official Review · Reviewer_U8cN · 2024-11-09
**Somewhat interesting lower bound results; very poor writing.**

**Rating:** 5
**Confidence:** 3

**Review:**

**Summary**: It is known that for noiseless sparse problems, rotation-invariant algorithms perform significantly worse than non-rotation invariant algorithms.  This paper shows that this qualitative statement holds true even when the sparse problems have noise.  Part of the reason why this is interesting is that these new lower bounds hold for the case where the sample size n exceeds the input dimension d  (whereas in the noiseless case, the sample size n is necessarily smaller than the input dimenion d).  The high-level idea to build such lower bounds is to first rotationally symmetrize the problem, and then build lower bounds for this symmetrized problem.

**Recommendation**: Based on the strengths and weaknesses outlined below, I lean towards rejecting the paper.  Even though the main lower bound is interesting, it in itself seems insufficient for acceptance.  To supplement the result, the authors do provide some results on continuous time trajectories, but these are not especially interesting, and quite disjointed/unrelated to the rest of the paper.  In addition, the writing is quite poor.  This reviewer is knowledgeable in the optimization side of things, but less so on the statistical side --- as such his confidence score is on the low end.

**Strengths**:
1. The main result (that rotation-invariant algorithms suffer on noisy sparse problems) is interesting, especially because in this case the sample size can exceed the input dimension.  To the author's knowledge, the proof of the lower bound seems correct.  The high-level idea of first symmetrizing the problem, and then proving lower bounds for this symmetrized problem is very natural.  [[I am not sure if this approach has been used previously in the literature (it has certainly been used to prove optimization lower bounds).]]
2. Literature is cited appropriately, and ideas seem to be correctly attributed.

**Weaknesses**:
1. The main result confirms an existing high-level observation (rotation-invariant algorithms suffer); it does not propose or support a *new* observation.
2. This reviewer finds the results in Sec 3 (closed-forms for continuous-time trajectories) uninteresting, and also disjointed from the main story on lower bounds.  Moreover, this qualitative observation that certain continuous time algorithms spend time near sparse solutions is already very well-established.  Instead of having this section, it would be better if the authors could supplement their lower bound by answering some of the open questions laid out in Sec 5.  This would strengthen the paper.
3. The paper is very poorly written.  As just one early example, "prove much lower upper bounds" is confusing.  Something like "prove much better upper bounds" sounds better.

**Other small points**:
* Theorem 3 (equivalence of continuous-time mirror descent, Riemannian gradient descent, preconditioning) is folklore in the optimization community, and is not new.

**Questions**:
* It is known that for fully connected networks and symmetric initialization, gradient descent is rotation-invariant.  To build non-rotation invariant algorithms, is there a natural non-symmetric initialization one can use which makes fully-connected neural nets prefer sparse solutions?  For example, what if the intialization itself is sparse?

**Paper Award:**

No

---

> ### Author Response · Authors · 2024-11-21
> **Re the continuous trajectory curves**
>
> Note that the other reviewers found the analytic trajectory plots valuable.
> But yes, we can strengthen the connection between the trajectory section
> and the rest of the paper:
> - Add a plot that shows that the exact solutions we computed actually closely track the discrete iterative algorithms.
> - Add the curves for Adam to Fig 2 that show it also stays away from the sparse solution in the noisy over-constrained case (it closely tracks AdaGrad)
>
> Frankly, the fact that the most common 1st order optimization algorithms (AdaGrad & Adam) stay away from sparse targets in the noisy over-constrained case is quite surprising
> - They get ``fooled by noise’’ as well and in some cases get misled by noise even more than rotation invariant algorithms.
>
> In contrast, we show there is a whole group of other simple algorithms that are not fooled by noise with sparse targets, including vanilla Gradient Descent on the spindly network of Fig. 1

---

> > ### Comment · Reviewer_U8cN · 2024-11-27
> > **Increase score**
> >
> > Based on the reviewer's response, I increased my score to 5.  The extension to the noisy over-constrained case seems more important than I originally thought.

---

> ### Author Response · Authors · 2024-11-24
> **Initialization that handles sparsity?**
>
> Your comment:
> “Is there a natural non-symmetric initialization one can use which makes fully-connected neural nets prefer sparse solutions? For example, what if the initialization itself is sparse?”
>
> Great question!:
> There are of course trivial ways to do this, by initializing with sparse weights and setting the learning rate at or near zero. This would build a preference for a specific sparse solution, not a preference that encompassed all sparse solutions. We conjecture it is not possible to initialize a dense network to have a uniform preference for all sparse solutions, essentially because within this architecture (we conjecture) it is not possible to have permutation invariance without rotation invariance, but we must leave this for future work.
>
> See also the discussion at the beginning of Appendix G about how the lower bound can be beaten in a dense network if the input distribution is not symmetric.

---

### Official Review · Reviewer_1Wvk · 2024-11-09

**Rating:** 6
**Confidence:** 3

**Review:**

This paper delves into the limitations of rotation-invariant algorithms, especially for learning sparse linear problems, highlighting that these algorithms struggle to generalize well when noise is added to sparse targets. It establishes theoretical lower bounds on generalization error for rotation-invariant algorithms, contrasting them with more adaptable algorithms like those using multiplicative updates or certain regularization techniques (e.g., Lasso). Through analysis of gradient flow trajectories, the authors illustrate how rotation-invariant algorithms cannot prioritize sparse features effectively and tend to treat noise and signal equally. Experimental observations suggest that non-rotation-invariant structures (like the spindly network) better learn sparse noisy targets, hinting at the importance of symmetry-breaking mechanisms in neural network design.

For the remarks of this paper, I would like to say:

1. The paper provides rigorous lower bounds for rotation-invariant algorithms in sparse linear settings, extending the discussion to noisy contexts where these algorithms continue to underperform. This fills a crucial gap in theoretical machine learning literature by offering both upper and lower bounds, which makes this paper novel and the contribution solid. The proof method is particularly robust for analyzing algorithm performance across various problem rotations

2. The paper is well written, and the reference list is complete. The analysis of the sparse linear model and the rotational invariant model contributes to the theoretical machine learning community. I would give this paper an accept.

3. I would like to say, in order to make it easier for the audience to understand, the authors can write more intuitive ideas and explanations for the theorems and lemmas, so that we can understand the full picture of the proofs at the first hand.

**Paper Award:**

No

---

### Official Review · Reviewer_S25Y · 2024-11-14

**Rating:** 7
**Confidence:** 3

**Review:**

It has been shown that rotation invariant algorithms are not optimal for sparse linear problems when $n<d$. This work shows that in the presence of added noise, the rotation invariant algorithms are sub-optimal even when $d<n$. This is done by establishing a lower bound for the Bayes optimal method for a rotationally symmetrized problem. The authors also show much better upper bounds for non-rotationally invariant for the same problem. Another interesting component is the analysis of gradient flow trajectories of popular optimization methods on the same problem.

Strengths: This is mostly a well-written and well-motivated paper. I found the findings to be very interesting and insightful. The analytic trajectories of the continuous-time versions of a variety of algorithms in section 3 are strong contributions. The Appendix contains detailed theoretical results about upper bounds on EG, approximate EGU algorithms, spindly network, and priming methods.

Weaknesses: The main weakness is that the authors assume much is already known to the reader. For someone who does not remember details of EG or EGU algorithms, it would be very useful to give some intuition or remind the reader why some of these are non-rotationally invariant.

Questions/suggestions:

1)	Is it possible to move some of the theorems from the appendix to the main paper? I understand the need for brevity under space constraints, but I feel that even if one of two important theorems can be moved, it will give a more concrete picture of the differences between the errors.

2)	From Figure 2 c) it seems that EGU+/- gets bad as time grows. However, Theorem 8 shows the error upper bound only for T=Theta(sqrt(d)). I tried to follow through with the proof to see how much of it could be extended for a larger T but I was not successful. Some discussion on that (also for the other theorems) connecting to Figure 2c) would be useful.

3)	Is there a typo in Eq (7) inside the square root?

4)	There are many short forms, LS, LLS, RFM, EGU, GD, etc. While some are obvious, it will be better if the authors write the full form the first time any of them show up.

**Paper Award:**

No

---

### Author Rebuttal · Authors · 2024-11-24

Thanks for the thorough reviews & many good suggestions. We will
- move some of the proofs to the main section,
- give more intuition about the proofs

Typo in Figure 2c: Entire top row (including 2c) is for sparse targets

Reviewer 1:
“Give more intuition for EG, EGU”: They are motivated by a regularization with a relative entropy. This puts the gradient into the exponent. The exponential is element wise so it is not preserved under rotation (the exponential and rotation operations do not commute & algs are not rotation-invariant)

In Fig. 2c, “EGU± gets bad as time grows.” All updates are derived as tradeoffs of a regularization term with the loss. As the number of noisy examples increases, the influence of the regularization term decreases and all algs minimize the square loss & converge to the LLS solution. Early stopping is crucial, aiming to hit the bottom of the dips of Fig. 2c.

We will correct the typos and define the acronyms on their first appearance.
Eq 7 is correct but needs better formatting.

Reviewer 2:
We will add more intuitive explanations. For example, the intuition for Thm 1 is that a rotationally invariant algorithm must show the same performance on any rotated version of the problem. Thus we can bound the performance by a symmetrized problem in which the rotation is chosen at random and is unknown to the algorithm. Thm 2 simply applies this result to our regression task and proves a lower bound for the symmetrized problem by relating it to ridge regression

Reviewer 3:
“The main result confirms an existing high-level observation (rotation-invariant algorithms suffer)”:
The result was previously proved only for the noiseless under-constrained case but not the noisy over-constrained case. Other reviewers see this as filling a crucial gap in the theoretical literature

Re the trajectory plots:
  Note that the other reviewers found the analytic trajectory plots valuable. But we can integrate them better
and add curves for Adam to Fig 2 that show it also stays away from the sparse solution in the noisy over-constrained case.
The fact AdaGrad & Adam (the most common 1st order algs) stay away from sparse targets in the noisy over-constrained case is quite surprising and has broad implications for current ML practice
- In contrast, GD on the simple spindly net of Fig 1 is not fooled by noise

Agreed Thm 3 is not a breakthrough, but its formalization highlights the differences between rotationally invariant and noninvariant algorithms.

---

### Meta-Review · Area_Chair_PL1C · 2024-12-11

**Recommendation:** Accept
**Confidence:** 5

**Metareview:**

Reviewers overall found the paper interesting and insightful. The technical contribution was found to be significant, and while one reviewer had certain reservations those were addressed during the rebuttal hence all reviewers provided positive scores and are inclined towards acceptance.

Overall as an area chair I also found the paper interesting and seems to provide new ideas as well as new proof techniques that are interesting of their own rights and can potentially have future applications and impact.

**Paper Award:**

No